# Layer-diverse Negative Sampling for Graph Neural Networks

**Wei Duan**                                                                *wei.duan@student.uts.edu.au*
*Australian Artificial Intelligence Institute*
*University of Technology Sydney*

**Jie Lu**                                                                            *jie.lu@uts.edu.au*
*Australian Artificial Intelligence Institute*
*University of Technology Sydney*

**Yu Guang Wang**                                                     *yuguang.wang@sjtu.edu.cn*
*Institute of Natural Sciences*
*School of Mathematical Sciences*
*Shanghai Jiao Tong University*

**Junyu Xuan**                                                               *junyu.xuan@uts.edu.au*
*Australian Artificial Intelligence Institute*
*University of Technology Sydney*

**Reviewed on OpenReview:** *https://openreview.net/forum?id=WOrdoKbxh6*

## Abstract

Graph neural networks (GNNs) are a powerful solution for various structure learning applications due to their strong representation capabilities for graph data. However, traditional GNNs, relying on message-passing mechanisms that gather information exclusively from first-order neighbours (known as positive samples), can lead to issues such as over-smoothing and over-squashing. To mitigate these issues, we propose a layer-diverse negative sampling method for message-passing propagation. This method employs a sampling matrix within a determinantal point process, which transforms the candidate set into a space and selectively samples from this space to generate negative samples. To further enhance the diversity of the negative samples during each forward pass, we develop a space-squeezing method to achieve layer-wise diversity in multi-layer GNNs. Experiments on various real-world graph datasets demonstrate the effectiveness of our approach in improving the diversity of negative samples and overall learning performance. Moreover, adding negative samples dynamically changes the graph's topology, thus with the strong potential to improve the expressiveness of GNNs and reduce the risk of over-squashing.

## 1 Introduction

Graph neural networks (GNNs) have emerged as a formidable tool for various applications of structure learning, including drug discovery (Sun et al., 2020), recommendation systems (Yu & Qin, 2020), and traffic prediction (Lan et al., 2022), owing to their strong representation learning power. GNNs propagate the learning of nodes through a message-passing mechanism (Geerts et al., 2021) that conveys and aggregates information from neighbouring nodes, known as first-order neighbours. The message passing is based on the assumption that neighbours of a node have similar representations. The common practice of updating node representations solely with positive samples in most GNNs (Kipf & Welling, 2017; Xu et al., 2019; Brody et al., 2022), can have three limitations: 1) Over-smoothing (Chen et al., 2020; Rong et al., 2020; Zhao & Akoglu, 2020), where the node representations become less distinct as the number of layers increases; 2) GNNs expressivity (Xu et al., 2019), where it becomes difficult to distinguish different graph topologies after aggregation; and 3) Over-squashing (Alon & Yahav, 2021; Topping et al., 2022; Karhadkar et al., 2022), where bottlenecks exist and limit the information passing between weakly connected subgraphs.

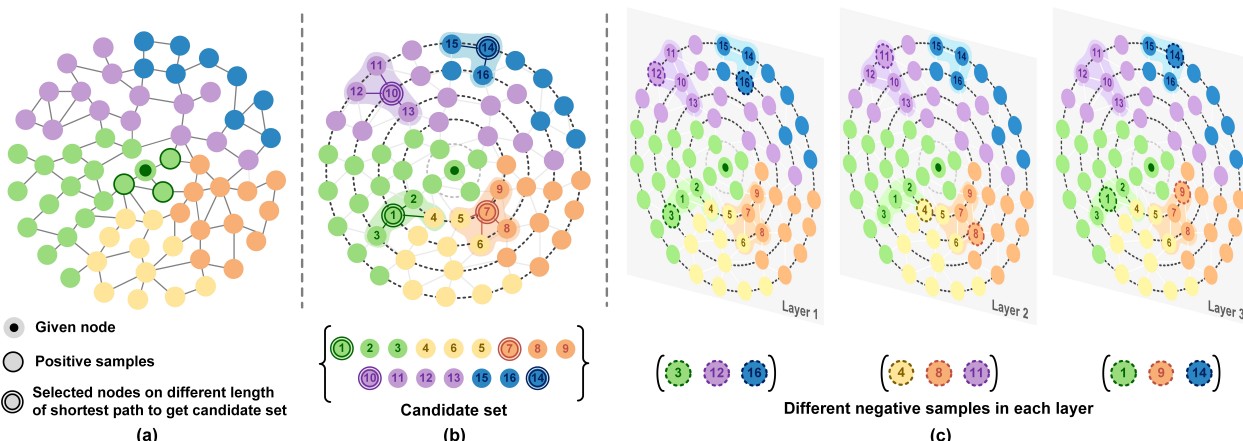

Figure 1: Negative samples from layer-diverse DPP sampling. (a) For a given node in a graph, its first-order neighbours can be thought of as *positive samples*, despite the fact that these neighbours may belong to different clusters. (b) Algorithm 1 calculates the shortest path from a given node to other nodes in the graph to obtain smaller, yet more efficient candidate sets for further sampling. (c) As the candidate set is significantly larger than the number of negative samples needed, the ideal subset of negative samples is not unique. By using the layer-diverse DPP sampling method to select negative samples, it is possible to include as much information from the entire graph as possible while also reducing redundancy among negative samples in different layers.

In addition to a node's positive samples, there are many other non-neighbouring nodes that can provide diverse and valuable information for updating the representations. Unlike neighbouring nodes, non-neighbouring nodes typically have distinct representations compared to the given node and are referred to as *negative samples* (Duan et al., 2022). While it is crucial to select appropriate negative samples, only a few studies have given adequate attention to this aspect of negative sampling.

It is believed that the ideal negative samples should *contain enough information about the entire graph without including a large amount of redundant information*. However, a remaining issue is that all previous approaches treat the negative samples in each layer as independent. Thus, from a holistic perspective, the negative samples obtained still contain a considerable amount of redundancy. In fact, experiments show that the overlap between the node samples for different layers obtained by Duan et al. (2022) is more than 75%, as outlined in further detail in Section 4.2.

To address the issue of redundant information in negative samples, we propose an approach called *layer-diverse negative sampling* that utilizes the technique of *space squeezing*. This method is designed to obtain meaningful information with a smaller number of samples (as illustrated by Figure 1). Specifically, our method utilizes the sampling matrix in DPP to transform the candidate set into a space. The dimension of the space is the number of nodes in the candidate set, with each node represented as a vector in this space. We then apply the space squeezing technique during sampling, which eliminates the dimensions corresponding to the samples of the last layer, thus significantly reducing the probability of selecting those samples again. These negative samples are then utilized in message passing in GCN, resulting in a new model called Layer-diverse GCN, or LDGCN in short.

The effectiveness of the LDGCN model has been demonstrated through extensive experimentation on seven publicly available benchmark datasets. These experiments have shown that LDGCN consistently achieves excellent performance across all datasets. We provide a detailed discussion on why the use of layer-diverse DPP sampling for negative samples may improve learning ability by improving GNNs expressivity and reducing the risk of over-squashing. Our main contributions are twofold:

- We propose a method for layer-diverse negative sampling that utilizes space squeezing to effectively reduce redundancy in the samples found and enhance the overall performance of the model.

- We empirically demonstrate the effectiveness of the proposed method in enhancing the diversity of layer-wise negative samples and overall node representation learning performance, and we also show the great potential of negative samples in improving GNNs expressivity and reducing the risk of over-squashing.

## 2 Preliminaries

### 2.1 Determinantal Point Processes (DPP)

A determinantal point process (DPP) $\Psi$ is a probability measure on all possible subsets of the ground set $\mathbb{Y}$ with a size of $2^{|\mathbb{Y}|}$. For every subset $\mathbb{Y}_{\text{sub}} \subseteq \mathbb{Y}$, a DPP (Hough et al., 2009) defined via a positive semidefinite $\boldsymbol{L}$ matrix is formulated as

$$\Psi_{\boldsymbol{L}}(\mathbb{Y}_{\text{sub}}) = \frac{\det\left(\boldsymbol{L}_{\mathbb{Y}_{\text{sub}}}\right)}{\det(\boldsymbol{L} \pm \boldsymbol{I})}, \tag{1}$$

where $\det(\cdot)$ denotes the determinant of a given matrix, $\boldsymbol{L}$ is a real and symmetric $|\mathbb{Y}| \times |\mathbb{Y}|$ matrix indexed by the elements of $\mathbb{Y}$, and $\det(\boldsymbol{L} + \boldsymbol{I})$ is a normalisation term that is constant once the ground dataset $\mathbb{Y}$ is fixed.

DPP has an intuitive geometric interpretation. If we have a $\boldsymbol{L}$, there is always a matrix $\boldsymbol{B}$ that satisfies $\boldsymbol{L} = \boldsymbol{B}^\top \boldsymbol{B}$. Let $\boldsymbol{B}_i$ be the columns of $\boldsymbol{B}$. A determinantal operator can then be interpreted geometrically as

$$\Psi_{\boldsymbol{L}}(\mathbb{Y}_{\text{sub}}) \propto \det\left(\boldsymbol{L}_{\mathbb{Y}_{\text{sub}}}\right) = \text{vol}^2\left(\{\boldsymbol{B}_i\}_{i \in \mathbb{Y}_{\text{sub}}}\right), \tag{2}$$

where the right-hand side of the equation is the squared $|\mathbb{Y}_{\text{sub}}|$-dimensional volume of the parallelepiped spanned by the columns of $\boldsymbol{B}$ corresponding to the elements in $\mathbb{Y}_{\text{sub}}$. Intuitively, diverse sets are more probable because their feature vectors are more orthogonal and span larger volumes. (See the more details in Appendix A.1)

### 2.2 Negative Sampling for GNNs

Let $\mathcal{G} = (\mathbb{V}, \mathbb{E})$ denote a graph with node features $\boldsymbol{h}_i$ for $i \in \mathbb{V}$, where $\mathbb{V}$ and $\mathbb{E}$ are the sets of nodes and edges. Let $N := |\mathbb{V}|$ denote the number of nodes. GNNs aggregate information via message-passing (Geerts et al., 2021), where each node $i$ repeatedly receives information from its first-order neighbours $\mathbb{N}_i$ to update its representation as

$$\boldsymbol{h}_i{}^l = \sum_{j \in \mathbb{N}_i \cup \{i\}} \frac{1}{\sqrt{\deg(i)} \cdot \sqrt{\deg(j)}} \left(\boldsymbol{w}^l \cdot \boldsymbol{h}_j^{(l-1)}\right), \tag{3}$$

where $\deg(\cdot)$ is the degree of the node. Introducing negative samples can improve the quality of the node representations and alleviate the over-smoothing problem (Chen et al., 2020; Rong et al., 2020; Duan et al., 2022). The new update to the representations is formulated as

$$\boldsymbol{h}_i{}^l = \sum_{j \in \mathbb{N}_i \cup \{i\}} \frac{1}{\sqrt{\deg(i)} \cdot \sqrt{\deg(j)}} \left(\boldsymbol{w}^l \cdot \boldsymbol{h}_j^{(l-1)}\right) - \mu \sum_{\bar{j} \in \overline{\mathbb{N}}_i} \frac{1}{\sqrt{\deg(i)} \cdot \sqrt{\deg(\bar{j})}} \left(\boldsymbol{w}^l \cdot \boldsymbol{h}_{\bar{j}}^{(l-1)}\right), \tag{4}$$

where $\overline{\mathbb{N}}_i$ are the negative samples of node $i$, and $\mu$ is a hyper-parameter to balance the contribution of negative samples.

The current state-of-the-art approaches for selecting negative samples $\overline{\mathbb{N}}_i$ used in Eq. (4) are the DPP-based methods (Duan et al., 2022; 2023). Intuitively, good negative samples for a node should have different semantics while containing as complete knowledge of the whole graph as possible. Since the sampling procedure in the DPP requires an eigendecomposition, the computational cost for once sampling from a $N$

---

**Algorithm 1:** Get candidate set $\mathbb{S}_i$ using shortest-path-based method

---

**Input** : A graph $\mathcal{G}$, sample length $P$, node $i$

**1** Compute the shortest path lengths from $i$ to all reachable nodes $\mathcal{V}_r$;
**2** Divide $\mathcal{V}_r$ into different sets $\mathbb{V}_p$ based on the path length $p$;
**3** $\mathbb{S}_i \leftarrow \emptyset$;
**4** **for** $p \leftarrow 2$ **to** $P$ **do**
**5** $\quad$ Randomly choose a node $j$ in $\mathbb{V}_p$;
**6** $\quad$ Collect first-order neighbours $\mathbb{N}_j$ of $j$;
**7** $\quad$ $\mathbb{S}_i \leftarrow \mathbb{S}_i \cup \mathbb{N}_j \cup j$;
**8** **end**

**Output:** Candidate set $\mathbb{S}_i$

---

nodes set could be $O(|N|^3)$, and the total becomes an excessive $O(|N|^4)$ for sampling $N$ times (all nodes). Thus, the large size of candidates found from exploring the whole graph to find negative samples would make such an approach impractical, even for a moderately sized graph.

To reduce the computational complexity, the shortest-path-based method (Duan et al., 2023) is first used to form a smaller but more effective candidate set $\mathbb{S}_i$ for node $i$, which is detailed in Algorithm 1. Using this method, the computational cost is approximately $O((P \cdot \overline{\deg})^3)$, where $P \ll N$ is the path length (normally smaller than the diameter of the graph) and $\overline{\deg} \ll N$ is the average degree of the graph. As an example, consider a Citeseer graph (Sen et al., 2008) with 3,327 nodes and $\overline{\deg} = 2.74$. When using the experimental setting shown below, where $P = 5$, we observe that $O(N^3) = 3.6 \times 10^{10}$, which is significantly larger than $2571 = O((P \cdot \overline{\deg})^3)$.

After obtaining the candidate set $\mathbb{S}_i$, the subsequent step involves effectively leveraging the characteristics of the graph to devise the computation method for the $\boldsymbol{L}$ matrix. As a major fundamental technique of DPP, Quality-Diversity decomposition is used to balance the diversity against some underlying preferences for different items in $\mathbb{Y}$ (Kulesza & Taskar, 2012). Since $\boldsymbol{L}$ can be written as $\boldsymbol{L} = \boldsymbol{B}^\top \boldsymbol{B}$, each column of $\boldsymbol{B}$ is further written as the product of a **quality** term $\boldsymbol{q}_{\bar{j}} \in \mathbb{R}^+$ and a vector of normalized **diversity** features $\boldsymbol{\phi}_{\bar{j}} \in \mathbb{R}^D, \|\boldsymbol{\phi}_{\bar{j}}\| = 1$. The probability of a subset is the square of the volume spanned by $\boldsymbol{q}_{\bar{j}}\boldsymbol{\phi}_{\bar{j}}$ for $\bar{j} \in \mathbb{Y}$. Hence, $\boldsymbol{L}$ for the given node $i$ now becomes

$$\boldsymbol{L}^i_{\bar{j}\bar{j}'} = \boldsymbol{q}^i_{\bar{j}}\boldsymbol{\phi}^\top_{\bar{j}} \boldsymbol{\phi}_{\bar{j}'}\boldsymbol{q}^i_{\bar{j}'}, \tag{5}$$

where $\bar{j}, \bar{j}' \in \mathbb{S}_i$ are two candidate negative nodes.

Utilizing the Quality-Diversity decomposition of DPP, we use the node feature representations and graph structural information to calculate $\boldsymbol{L}$ for the intended sample selection. Specifically, following the Duan et al. (2023), all the nodes $\mathbb{V}$ in a graph $\mathbb{G}$ are first divide into $Q$ communities, denoted as $\mathbb{V} = \left\{\mathbb{V}^{com}_q\right\}^Q_{q=1}$, using Fluid Communities method (Parés et al., 2017). Then, the features of each community $\mathbb{V}^{com}_q$ and each candidate set $\mathbb{S}_i$ are extracted from the node representations $\boldsymbol{h}_i$ via

$$\boldsymbol{a}_q = \frac{\sum_{i \in \mathbb{V}^{com}_q} \boldsymbol{h}_i}{\left|\mathbb{V}^{com}_q\right|}, \quad \boldsymbol{b}_i = \frac{\sum_{\bar{j} \in \mathbb{S}_i} \boldsymbol{h}_{\bar{j}}}{|\mathbb{S}_i|}. \tag{6}$$

With the Eq. (6), the **quality terms** in Eq. (5) will be defined as:

$$\boldsymbol{q}^i_{\bar{j}} = \cos(\boldsymbol{a}_i, \boldsymbol{b}_i) \odot \cos(\boldsymbol{a}_i, \boldsymbol{a}_{\bar{j}}), \quad \boldsymbol{q}^i_{\bar{j}'} = \cos(\boldsymbol{a}_i, \boldsymbol{b}_i) \odot \cos(\boldsymbol{a}_i, \boldsymbol{a}_{\bar{j}'}), \tag{7}$$

where $\boldsymbol{a}_i, \boldsymbol{a}_{\bar{j}}, \boldsymbol{a}_{\bar{j}'}$ represents the feature expression of the node belonging to its community, $\boldsymbol{b}_i$ denotes the features of the candidate set $\mathbb{S}_i$ and the $\odot$ means point-wise product. These **quality terms** ensure the candidate node $\bar{j}$ is not similar to the given node $i$. The **diversity term** in Eq. (5) is defined as

$$\boldsymbol{\phi}^\top_{\bar{j}} \boldsymbol{\phi}_{\bar{j}'} = \cos(\boldsymbol{h}_{\bar{j}}, \boldsymbol{a}_{\bar{j}'}) \cos(\boldsymbol{a}_{\bar{j}}, \boldsymbol{h}_{\bar{j}'}) \odot \exp\left(\cos((\boldsymbol{h}_{\bar{j}}, \boldsymbol{h}_{\bar{j}'}) - 1)\right), \tag{8}$$

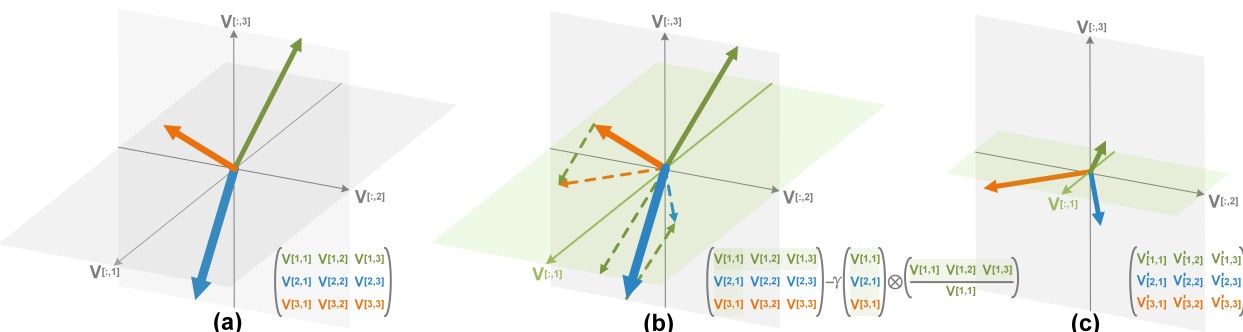

Figure 2: Illustration of the layer-diverse sampling process. (a) In the candidate set with 3 nodes, construct the $\boldsymbol{V}^{3\times 3}$. The original space is spanned by the eigenvectors $\boldsymbol{v}_1, \boldsymbol{v}_2, \boldsymbol{v}_3$ and every node in the candidate set corresponds to a coloured vector in this space. (b) Suppose node 1 (green vector) is selected in the last layer, which has the greatest impact on the $\boldsymbol{v}_1/\boldsymbol{V}[:, 1]$, we then squeeze the space along the $\boldsymbol{V}[:, 1]$ direction. If the sign of another node in $\boldsymbol{V}[:, 1]$ projection is the same as the green one, the re-scale direction will be the same (the orange vector) and vice versa (the blue vector). (c) This operation will result in a new space, where the component $\boldsymbol{V}[:, 1]$ is significantly cut-off, which means the probability of picking the corresponding node 1 has been reduced.

which ensures that there are sufficient differences between every pair of candidate nodes $\bar{j}$ and $\bar{j}'$.

Due to the primary focus of this paper not being on the computation of the $\boldsymbol{L}$ matrix, additional details can be referenced in Duan et al. (2022; 2023). Although the above method ensures good diversity for each layer, there is still plenty of redundancy across the layers because the negative samples in each layer are treated as being independent.

## 3 Proposed Model

### 3.1 Layer-diverse Negative Sampling

Given a node $i$ and its candidate negative sample set $\mathbb{S}_i{}^1$. $\overline{\mathbb{N}}_i^l \in \mathbb{S}_i$ and $\overline{\mathbb{N}}_i^{l+1} \in \mathbb{S}_i$ denote the negative samples for node $i$ in a multi-layer GNNs collected from layers $l$ and $l+1$, respectively, as illustrated in Figure 1. Our goal is to reduce the overlap between $\overline{\mathbb{N}}_i^l$ and $\overline{\mathbb{N}}_i^{l+1}$ to cover as much negative information in the graph as possible while retaining an accurate representation of $i$. Note that $\mathbb{S}_i$ could be seen geometrically as a space spanned by the node representations, while $\overline{\mathbb{N}}_i^l$ is just a subspace of this space spanned by the selected negative samples/representations. Inspired by this geometric interpretation of negative sampling, our idea is to squeeze this space for negative sampling at layer $l+1$ conditioned on obtained $\overline{\mathbb{N}}_i^l$ to reduce the probability of re-picking the samples in $\overline{\mathbb{N}}_i^l$.

To be specific, an eigendecomposition is first performed on $\boldsymbol{L}$ from Eq. (5) of $\mathbb{S}_i = \{\bar{j}_1, ..., \bar{j}_S\}$, which yields the eigenvalues $\mathbb{U} = \{\lambda_1, ..., \lambda_S\}$ and the eigenvectors $\mathbb{T} = \{\boldsymbol{v}_1, ..., \boldsymbol{v}_S\}$. Here, $\mathbb{T}$ is an orthogonal basis for the space of $\mathbb{S}_i$ since $\boldsymbol{L}$ is a real and symmetric matrix. All the eigenvectors compose a new matrix denoted as $\boldsymbol{V}^{S\times S} = [\boldsymbol{v}_1, ..., \boldsymbol{v}_S]$, where each row corresponds to a node in $\mathbb{S}_i$, and $\boldsymbol{V}[\bar{j}, :]$ is also the impacts/contributions of the node $\bar{j}$ on each eigenvector. The probability of picking node $\bar{j}$ through the DPP sampling is then proportional to $||\boldsymbol{V}[\bar{j}, :]||_2$. Given $\overline{\mathbb{N}}_i^l$, the goal is reduce the information of any $\bar{j}^* \in \overline{\mathbb{N}}_i^l$ in space $\boldsymbol{V}$. To this end, the eigenvector/basis of node $\bar{j}^*$ that makes the greatest impact/contribution is identified as:

$$m = \arg \max_{y\in\{1,2,..,S\}} \boldsymbol{V}[\bar{j}^*, y].$$
(9)

---

[1]$\mathbb{S}_i$ denotes all non-neighbour nodes of $i$ in the graph in theory, but we follow Duan et al. (2022; 2023) to reduce its size by the Algorithm 1.

---

**Algorithm 2:** Layer-diverse negative sampling

**Input**  : Node $i$, candidate set $\mathbb{S}_i, \overline{\mathbb{N}}_i^{l-1}$

**1** Calculate $\boldsymbol{L}^i$ using Eq. (5);

**2** Eigendecompose $\boldsymbol{L}^i$ to get $\mathbb{U}$ and $\mathbb{T}$;

**3 for** *every* $\bar{j}^* \in \overline{\mathbb{N}}_i^{l-1}$ **do**

**4**     Find $m$ using Eq. (9);

**5**     Get layer-diverse matrix $\mathbb{V}'$ using Eq. (10);

**6 end**

**7** Perform $k$-DPP sampling on $\mathbb{S}_i$ using Algorithm 3;

**Output:** $\overline{\mathbb{N}}_i^l$

---

The space $\boldsymbol{V}$ along the $m$ direction is then squeezed by

$$\boldsymbol{V}' = \boldsymbol{V} - \gamma \boldsymbol{V}[:, m] \otimes \frac{\boldsymbol{V}[\bar{j}^*, :]}{\boldsymbol{V}[\bar{j}^*, m]}, \tag{10}$$

where $\otimes$ denotes the outer product and $\gamma \in (0, 1)$ is the weight of the squeezing. The outer product can be thought of as a way to "stretch" every vector of node $\bar{j}^*$ along the $\boldsymbol{V}[:, m]$-direction. Since $m$ in Eq. (10) implies that the node $\bar{j}^*$ has the strongest influence on this eigenvector/direction, it helps to reduce the contribution of the node $\bar{j}^*$ at $m$-direction to all vectors as much as possible. It is worth noting about Eq. (10) that:

**Remark 3.1.** *Suppose the probability of re-picking node $\bar{j}^* \in \overline{\mathbb{N}}_i^l$ in $\boldsymbol{V}$ is p, the new probability of re-picking it in $\boldsymbol{V}'$ would be reduced to $(1 - \gamma)p$, where $0 \leq \gamma \leq 1$. It means that we can control the squeezing degree by $\gamma$. See the proof given in Appendix A.2.*

**Remark 3.2.** *For a node $\bar{i} \in \overline{\mathbb{N}}_i^l$ and $\bar{i} \neq \bar{j}^*$, if $\boldsymbol{V}[\bar{i}, :]$ and $\boldsymbol{V}[\bar{j}^*, :]$ are sufficiently similar with each other, then the probability of re-picking $\bar{i}$ would also be reduced. (See the proof in Appendix A.2.) It means we do not just reduce the re-picking probability of $\bar{j}^*$. By reducing the re-picking probability of $\bar{j}^*$, we also decrease the influence of similar nodes, reducing the likelihood of them being considered.*

After obtaining the layer-diverse vector matrix $\boldsymbol{V}'$, we employ $k$-DPP for negative sampling. $k$-DPP is a generalization of the DPP for sampling a fixed number of items, rather than a variable number. By setting the value of $k$, we can effectively control the number of negative samples obtained through sampling. (See the more details in Appendix A.1) The process of layer-diverse negative sampling for node $i$ is outlined in Algorithm 2. The most significant difference lies in the original input of $k$-DPP is eigenvector matrix $\boldsymbol{V}$, while the input of Algorithm 3 is layer-diverse matrix $\boldsymbol{V}'$. Our method can be used for all layers by collecting all negative samples before a given layer as the candidate set. To reduce the computational cost, two consecutive layers are used in the following experiments.

To better illustrate our method, a three-dimensional example is shown in Figure 2, where the candidate set contains three nodes and the size of $\boldsymbol{V}$ is $3 \times 3$. Figure 2(a) shows that the original space is spanned by $\boldsymbol{v}_1, \boldsymbol{v}_2, \boldsymbol{v}_3$, with the eigenvectors $\{\boldsymbol{V}[:, y]\}_{y=1,2,3}$. Suppose node 1 has the highest impact on $\boldsymbol{v}_1$, that is $1 = \arg\max \boldsymbol{V}[:, 1]$. The space along the $\boldsymbol{v}_1$-direction then squeezes, as we can observe in Figure 2(b). The original space finally turns to the new space in Figure 2(c), where the magnitude of the $\boldsymbol{v}_1$ component is significantly reduced and the probability of choosing the corresponding nodes (including node 1) becomes smaller.

## 3.2 Discussion

Although there is a limited number of works having investigated the use of negative samples for GNNs, the exact benefits of using such samples remain largely unexplored. To better understand the impact of negative samples, we give examples and discussions on their effects on GNNs expressivity and over-squashing. Our

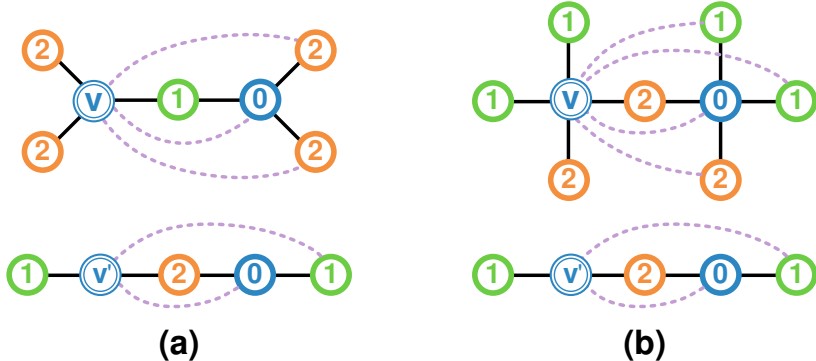

Figure 3: Case 1: Adding negative samples can help GNN learn different embedding for different structures. Dash lines mean adding negative samples. (a) After adding negative samples, MAX can distinguish different structures. (b) After adding negative samples, MAX and MEAN can distinguish different structures.

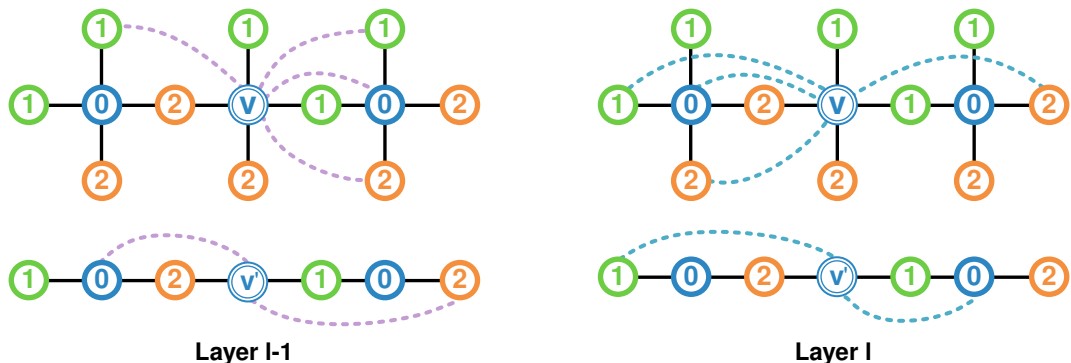

Figure 4: Case 2: Although for layer $l-1$, MAX and MEAN aggregators still can not distinguish different structures after adding negative samples. Since the layer-diverse method can obtain different samples from the last layer, for layer $l$, adding negative samples lets MAX and MEAN aggregators to be able to distinguish different structures.

results show that negative samples have a strong potential to improve GNNs expressivity and reduce the degree of over-squashing.

### 3.2.1 GNN expressivity

Intuitively, adding negative samples into a graph's convolution layers will temporarily change the graph's topologies. Xu et al. (2019) state that ideally powerful GNNs can distinguish between different graph structures by mapping them into different embeddings (so-called *GNN expressivity*). In the following, from a topological view, we will demonstrate the three different aggregation cases in which adding negative samples to GNNs may improve the GNN expressivity.

**Case 1** In a single layer, negative samples can help aggregators distinguish different structures. As shown in Figure 3(a), before adding negative samples, MAX fails to distinguish two structures because

$$v = \max(2, 2, 1) = \max(2, 1) = v'. \tag{11}$$

After adding negative samples, we have

$$v = 2 - \mu \max(2, 2, 0) \neq 2 - \mu \max(0, 1) = v'. \tag{12}$$

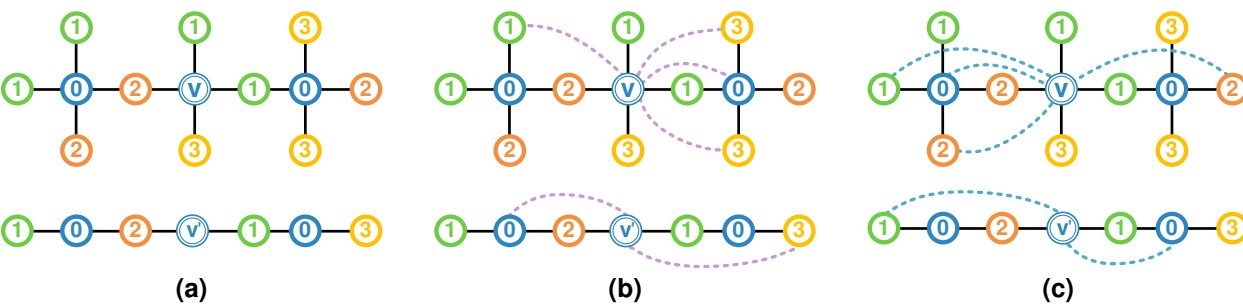

Figure 5: Case 3: (a) Aggregators in the original graph can distinguish different structures. (b) Under the specific condition, adding negative samples has a small probability of preventing that. (c) Even if this situation occurs, the layer-diverse approach will address this in the next layer.

In Figure 3(b), before adding negative samples, MAX and MEAN both fail to distinguish two structures because for MAX, it will be

$$v = \max(1, 1, 2, 2) = \max(2, 1) = v', \tag{13}$$

and for MEAN, it will be

$$v = \frac{1 + 1 + 2 + 2}{4} = \frac{1 + 2}{2} = v'. \tag{14}$$

After adding negative samples, we have

$$v = 2 - \mu \max(1, 1, 2, 0) \neq 2 - \mu \max(0, 1) = v'. \tag{15}$$

$$v = \frac{3}{2} - \mu \frac{1 + 1 + 0 + 2}{4} \neq \frac{3}{2} - \mu \frac{0 + 1}{2} = v'. \tag{16}$$

Negative samples help MAX and MEAN distinguish different structures in this case.

**Case 2** Layer-diverse negative samples can help distinguish different structures in multi-layers. The outcomes of only one negative sampling process do not always help the aggregators to generate different embeddings for various structures. As Figure 4 shows, even after adding some negative samples, the MAX and MEAN aggregators for layer $l - 1$ still cannot distinguish between the different structures. This is because we have:

$$v = \max(1, 1, 2, 2) - \mu \max(1, 1, 0, 2) = \max(1, 2) - \mu \max(0, 2) = v', \tag{17}$$

$$v = \frac{1 + 1 + 2 + 2}{4} - \mu \frac{1 + 1 + 0 + 2}{4} = \frac{1 + 2}{2} - \mu \frac{0 + 2}{2} = v'. \tag{18}$$

However, if the sampling method is well-designed, the probability of distinguishing between different graph structures in the network will be higher. Benefit from the layer-diverse method which can obtain different samples from the last layer, for layer $l$, we get different samples and have

$$v = \max(1, 1, 2, 2) - \mu \max(1, 0, 2, 2) \neq \max(1, 2) - \mu \max(1, 0) = v', \tag{19}$$

$$v = \frac{1 + 1 + 2 + 2}{4} - \mu \frac{1 + 0 + 2 + 2}{4} \neq \frac{1 + 2}{2} - \mu \frac{1 + 0}{2} = v'. \tag{20}$$

In this case, the layer-diverse negative sampling will help MAX and MEAN to distinguish different structures in multi-layers.

**Case 3** There exist some situations where adding negative samples could make the originally distinguishable structures indistinguishable, but the probability of such situations is low. As shown in Figure 5(a), the MAX and MEAN aggregators can distinguish two different structures because we have

$$v = \max(1, 1, 2, 3) \neq \max(1, 2) = v', \tag{21}$$

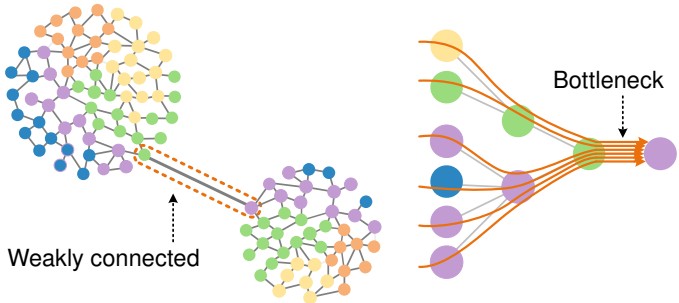

Figure 6: Over-squashing occurs when information passes between weakly connected subgraphs, where bottlenecks exist and lead to the graph failing to propagate messages flowing from distant nodes.

$$v = \frac{1+1+2+3}{4} - \neq \frac{1+2}{2} = v'. \tag{22}$$

As shown in Figure 5(b), in the $l-1$ layer, after adding negative samples, using MEAN will have

$$v = \frac{1+1+2+3}{4} - \mu \frac{1+3+0+3}{4} = \frac{7}{4} - \mu \frac{7}{4}. \tag{23}$$

$$v' = \frac{1+2}{2} - \mu \frac{0+3}{2} = \frac{3}{2} - \mu \frac{3}{2}. \tag{24}$$

We notice that it cannot distinguish two structures after adding negative samples when $\mu = 1$. However, from this example, we can see that to make the originally distinguishable structures indistinguishable, the negative samples and $\mu$ must exactly complement the difference between the two structures from the original aggregation. Apparently, the probability of finding satisfied the negative samples and $\mu$ is significantly smaller than the finding of some negative samples to make representations of two structures different. Furthermore, considering our layer-diverse design, the probability of finding such satisfied negative samples and $\mu$ at every layer would be exponentially reduced. As shown in Figure 5(c), the result of the MEAN operator is

$$v = \frac{7}{4} - 1 \times \frac{1+0+2+2}{4} \neq \frac{3}{2} - 1 \times \frac{0+1}{2} = v'. \tag{25}$$

Hence, we believe our layer-diverse negative sampling is helpful in improving GNN expressivity.

### 3.2.2 Over-smoothing

Separate from over-smoothing and GNN expressivity, over-squashing is much less known, first pointed out by Alon & Yahav (2021). Over-squashing occurs when bottlenecks exist and limit the information passing between weakly connected subgraphs, which leads to the graph failing to propagate messages flowing from distant nodes (Alon & Yahav, 2021; Topping et al., 2022), as shown in Figure 6. An effective approach to addressing over-squashing is to *rewire* the input graph to remove the structural bottlenecks Karhadkar et al. (2022). However, the rewiring methods face two main challenges: 1) losing the original topological information when the graph changes and 2) suffering from over-smoothing when adding too many edges. In the following, we will show negative samples have the potential to address these two challenges.

An alternative way to understand negative samples in Eq. (4) is to introduce new (negative) edges/relations into GNNs, which can be rewritten as

$$\boldsymbol{h}_i^l = \boldsymbol{w}^l \boldsymbol{h}_i^{(l-1)} + \sum_{(j,i)\in\mathbb{E}_1} \frac{1}{\boldsymbol{C}_{j,i}} \boldsymbol{w}_1^l \boldsymbol{h}_j^{(l-1)} + \sum_{(\bar{j},i)\in\mathbb{E}_2} \frac{1}{\boldsymbol{C}_{\bar{j},i}} \boldsymbol{w}_2^l \boldsymbol{h}_{\bar{j}}^{(l-1)}, \tag{26}$$

where $\mathbb{E}_1$ and $\mathbb{E}_2$ denote positive and negative relations separately. Firstly, as stated in Karhadkar et al. (2022), the flexible $\boldsymbol{w}_1$ and $\boldsymbol{w}_2$ could help to balance the over-smoothing and over-squashing. Secondly,

Table 1: Accuracy of all 4-layer models on datasets

|  | Citeseer | Cora | PubMed | CS | Computers | Photo | ogbn-arxiv |
|---|---|---|---|---|---|---|---|
| GCN | $55.78_{\pm 5.69}$ | $63.39_{\pm 7.92}$ | $72.24_{\pm 4.34}$ | $54.00_{\pm 3.69}$ | $47.21_{\pm 6.22}$ | $68.04_{\pm 6.37}$ | $70.57_{\pm 1.02}$ |
| GATv2 | $63.67_{\pm 7.07}$ | $74.43_{\pm 3.80}$ | $74.95_{\pm 1.71}$ | $85.00_{\pm 1.55}$ | $61.90_{\pm 5.38}$ | $79.08_{\pm 3.43}$ | $70.60_{\pm 0.86}$ |
| SAGE | $59.70_{\pm 8.87}$ | $73.13_{\pm 3.54}$ | $75.48_{\pm 1.94}$ | $82.22_{\pm 2.60}$ | $59.27_{\pm 7.85}$ | $79.01_{\pm 6.54}$ | $71.15_{\pm 1.00}$ |
| GIN-$\epsilon$ | $60.89_{\pm 1.97}$ | $68.07_{\pm 8.87}$ | $72.93_{\pm 5.09}$ | $59.00_{\pm 9.52}$ | $37.09_{\pm 2.21}$ | $31.56_{\pm 6.91}$ | $35.04_{\pm 5.33}$ |
| AERO | $62.35_{\pm 4.88}$ | $73.37_{\pm 6.83}$ | $72.80_{\pm 3.50}$ | $64.50_{\pm 15.70}$ | $50.20_{\pm 10.0}$ | $56.61_{\pm 14.54}$ | $70.04_{\pm 0.91}$ |
| RGCN | $62.82_{\pm 3.84}$ | $71.75_{\pm 3.64}$ | $74.96_{\pm 1.40}$ | $79.91_{\pm 3.50}$ | $56.44_{\pm 9.78}$ | $75.19_{\pm 8.60}$ | $71.19_{\pm 0.42}$ |
| MCGCN | $50.90_{\pm 9.70}$ | $69.28_{\pm 4.33}$ | $71.44_{\pm 4.09}$ | $80.66_{\pm 3.81}$ | $64.09_{\pm 7.27}$ | $73.01_{\pm 9.54}$ | $65.49_{\pm 0.26}$ |
| PGCN | $63.03_{\pm 4.87}$ | $70.37_{\pm 4.51}$ | $75.47_{\pm 1.78}$ | $52.73_{\pm 11.14}$ | $71.13_{\pm 6.27}$ | $79.26_{\pm 6.67}$ | $66.16_{\pm 0.45}$ |
| D2GCN | $63.30_{\pm 2.01}$ | $73.02_{\pm 3.01}$ | $75.36_{\pm 1.82}$ | $83.47_{\pm 2.94}$ | $74.19_{\pm 2,06}$ | $82.78_{\pm 4.23}$ | $71.46_{\pm 0.21}$ |
| **LDGCN** | $\mathbf{68.27}_{\pm 1.29}$ | $\mathbf{76.80}_{\pm 1.26}$ | $\mathbf{77.07}_{\pm 1.23}$ | $\mathbf{86.23}_{\pm 0.55}$ | $\mathbf{77.92}_{\pm 2.34}$ | $\mathbf{86.50}_{\pm 1.48}$ | $\mathbf{71.66}_{\pm 0.30}$ |

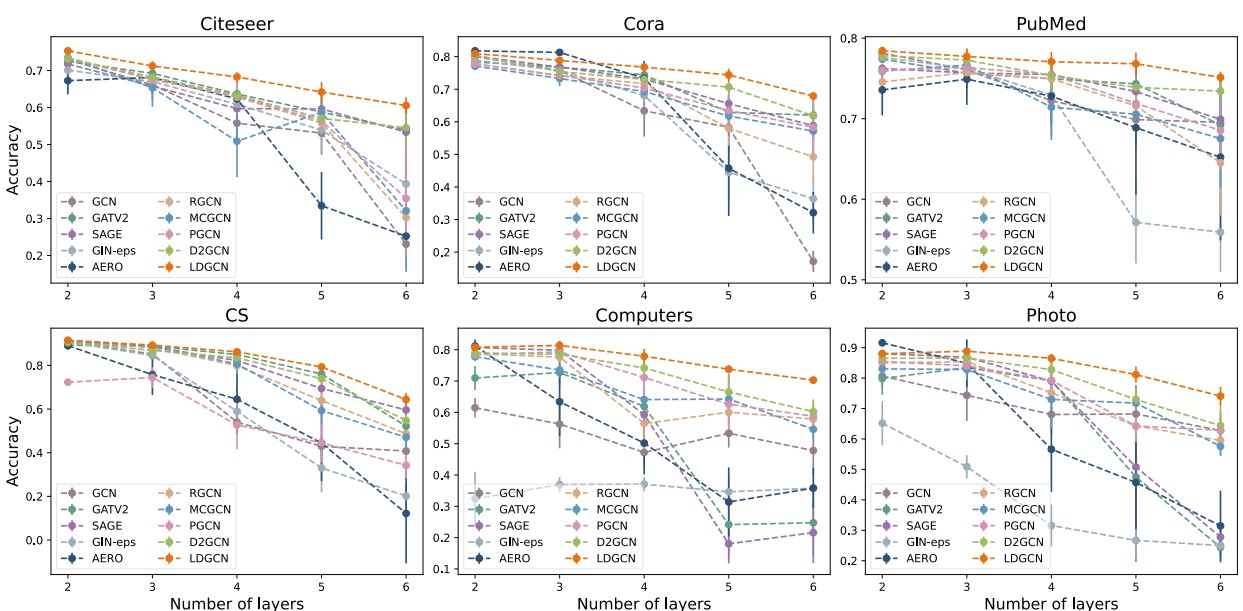

Figure 7: Node classification accuracy of all models with 2-6 layers in six datasets.

different from the positive samples/edges added by Karhadkar et al. (2022), our negative samples/edges could further improve the ability to preserve the node representations. The reason is that the underlying assumption of GNNs is that the representation of a node should be similar to the representations of its (positive) linked nodes, so any new positive samples might very likely bring some incorrect information to a node and then damage the original node representation significantly. However, our negative samples are purposely chosen to provide negative information to a given node, so the Eq. (4) would not damage the original node representation too much under the same number of newly added edges with Karhadkar et al. (2022). Hence, we believe the negative samples are useful to overcome the over-squashing problem.

## 4 Experiments

Our experiments aimed to address these questions: (1) Can the addition of negative samples obtained using our method improve the performance of GNNs compared to baseline methods? (Section 4.1) (2) Does our

method result in negative samples with reduced redundancy? (Section 4.2) (3) Does our method yield consistent results even when fewer nodes are included in the negative sampling? (Section 4.2) (4) How would our negative sampling approach perform when applied to other GNNs architectures? (Section 4.3) (5) Does incorporating these negative samples into graph convolution alleviate issues with over-smoothing and over-squashing? (Section 4.4) (6) What is the time complexity of the proposed method? (Section 4.5) (7) How do the sampling results of our LDGCN model compare to those of the D2GCN model in terms of overlap reduction and sample diversity? (Section 4.6)

## 4.1 Evaluation of Node Classification

**Datasets.** We first conducted our experiments with seven homophilous datasets for semi-supervised node classification, including citation network: **Citeseer**, **Cora** and **PubMed** (Sen et al., 2008), Coauthor networks: **CS** (Shchur et al., 2018), Amazon networks: **Computers** and **Photo** (Shchur et al., 2018), and Open Graph Benchmark: **ogbn-arxiv** (Hu et al., 2020). Then, we expanded our experiments to three heterophilous datasets, including **Cornell**, **Texas**, and **Wisconsin** Craven et al. (1998).

**Baselines.** For homophilous datasets, we compared our framework to four GNN baselines: GCN (Kipf & Welling, 2017), GATv2 (Brody et al., 2022), SAGE (Hamilton et al., 2017), GIN-$\epsilon$ (Xu et al., 2019), and AERO (Lee et al., 2023). We also compared existing GNN models with negative sampling methods. RGCN (Kim & Oh, 2021) selects negative samples in a purely random manner. MCGCN (Yang et al., 2020) selects negative samples using Monte Carlo chains. PGCN (Ying et al., 2018) uses personalised PageRank. D2GCN (Duan et al., 2022) calculates the $L$-ensemble using node representations only and does not take into account the diversity of the samples found across layers. Once the negative samples were obtained using these methods, they were integrated into the convolution operation using Eq. (4). For heterophilous datasets, to ensure a comprehensive analysis, we tested the layer-diverse negative sampling method across multiple graph neural network architectures, both in 2-layer and 4-layer configurations. The architectures tested include GCN (LD-GCN), GATv2 (LD-GATv2), GraphSAGE (LD-SAGE), and GIN (LD-GIN).

**Experimental setup.** We selected 1% of the nodes for negative sampling in each network layer. The datasets were divided consistently with Kipf & Welling (2017). Further information on the experimental setup and hyperparameters can be found in Appendix A.3.

**Results on homophilous datasets.** The results reported are the average accuracy values of the node classification after 10 runs, shown in Figure 7 for layers 2 to 6 and the detailed values for layer 4 are presented in Table 1. The results indicate that our model outperforms the other models. It performed better than the state-of-the-art GCN variants: GraphSAGE, GATv2, GIN-$\epsilon$, and AERO. Unlike these methods, LDGCN incorporates both neighbouring nodes (positive samples) and negative samples obtained by our method into message passing. Although RGCN, MCGCN, and PGCN also incorporate negative samples into the convolution operation, they have not shown consistent performance across various datasets. D2GCN does not utilize layer-diverse sampling to reduce the chances of selecting the same nodes in consecutive layers. Consequently, the negative samples identified by this method contain less information about the overall graph, impeding graph learning.

**Results on heterophilous datasets.** The results of the 2-layer are shown in Table 2. On the Cornell dataset, our LD-GCN model outperformed the standard GCN by approximately 6.84%. In Texas, the LD-GATv2 model showed an improvement of 11.32% over the standard GATv2. For Wisconsin, LD-SAGE exceeded the performance of standard GraphSAGE by 5.82%. Furthermore, the 4-layer model results (Table 3) are consistent with the improved performance observed in the 2-layer models, suggesting that our layer-diverse negative sampling method contributes positively across different model depths.

Heterophilous graphs are characterized by their tendency to connect nodes with dissimilar features or labels. This starkly contrasts the homophilous nature typically assumed in many GNN designs. This heterophily implies a diverse neighbourhood for each node, which can challenge learning algorithms that rely on the assumption that 'neighbouring nodes have similar labels or features'.

Our layer-diverse negative sampling method is well-suited for such graphs for several reasons:

Table 2: Acc of 2-layer models on WebKB dataset

|          | Cornell | Texas | Wisconsin |
|----------|---------|-------|-----------|
| GCN | $48.52_{\pm 5.09}$ | $56.21_{\pm 5.65}$ | $49.80_{\pm 5.70}$ |
| LD-GCN | $55.36_{\pm 6.04}$ | $61.62_{\pm 5.90}$ | $61.56_{\pm 5.63}$ |
| GATv2 | $51.35_{\pm 7.15}$ | $50.54_{\pm 4.21}$ | $50.54_{\pm 4.21}$ |
| LD-GATv2 | $66.48_{\pm 4.71}$ | $61.86_{\pm 7.36}$ | $64.31_{\pm 6.72}$ |
| GraphSAGE | $61.01_{\pm 4.17}$ | $70.27_{\pm 5.04}$ | $70.65_{\pm 2.86}$ |
| LD-SAGE | $67.11_{\pm 7.54}$ | $76.46_{\pm 4.52}$ | $76.47_{\pm 6.32}$ |
| GIN | $43.78_{\pm 4.49}$ | $56.48_{\pm 5.19}$ | $47.05_{\pm 5.33}$ |
| LD-GIN | $56.78_{\pm 3.05}$ | $61.56_{\pm 5.37}$ | $52.94_{\pm 4.16}$ |

Table 3: Acc of 4-layer models on WebKB dataset

|          | Cornell | Texas | Wisconsin |
|----------|---------|-------|-----------|
| GCN | $43.78_{\pm 6.70}$ | $54.23_{\pm 6.52}$ | $53.13_{\pm 6.92}$ |
| LD-GCN | $48.65_{\pm 5.37}$ | $58.64_{\pm 5.42}$ | $58.47_{\pm 6.02}$ |
| GATv2 | $49.54_{\pm 8.50}$ | $57.97_{\pm 6.33}$ | $51.52_{\pm 5.37}$ |
| LD-GATv2 | $52.54_{\pm 3.86}$ | $60.06_{\pm 3.78}$ | $57.14_{\pm 3.54}$ |
| GraphSAGE | $52.97_{\pm 6.41}$ | $64.86_{\pm 5.40}$ | $59.37_{\pm 5.28}$ |
| LD-SAGE | $58.59_{\pm 6.10}$ | $70.64_{\pm 7.61}$ | $62.94_{\pm 7.21}$ |
| GIN | $48.38_{\pm 7.29}$ | $58.39_{\pm 4.56}$ | $47.12_{\pm 4.43}$ |
| LD-GIN | $54.05_{\pm 4.69}$ | $60.48_{\pm 4.03}$ | $54.37_{\pm 3.15}$ |

- **DPP-based sampling within layers**: Our method uses DPP-based sampling to ensure diversity within each layer of the graph. This approach is crucial for heterophilous graphs, where it's important to capture a wide range of node characteristics within the same layer.

- **Layer-diverse enhancement**: We enhance diversity between layers and reduce overlap, allowing for richer information capture across the graph. This method is particularly effective in heterophilous graphs, where nodes with similar properties may not be close in the graph's topology.

- **Improved node representation learning:** Our approach effectively learns node representations by distinguishing between similar and dissimilar neighbors. This is key in heterophilous graphs, where traditional GNNs might struggle due to the uniformity in their aggregation and update processes.

- **Structural insight**: Our method offers more structural insight into the graph by allowing the GNN to learn from a wider range of node connections, thus avoiding the pitfall of homogeneity in the learning process.

We believe that these results and our analysis of the structural properties of heterophilous graphs demonstrate the applicability and advantages of our layer-diverse negative sampling method in a broader range of graph types. This strengthens the case for our approach as a versatile tool in the GNN toolkit, capable of addressing the challenges presented by both homophilous and heterophilous graphs.

Table 4: Overlap rates of D2GCN and LDGCN on Cora

| METHOD | $OVR_{node}$ | | $OVR_{cls}$ | | $OVR_{5\times cls}$ | |
|---|---|---|---|---|---|---|
| | 4-Layer | 6-Layer | 4-Layer | 6-Layer | 4-Layer | 6-Layer |
| D2GCN-1% | $75.00_{\pm 6.37}$ | $65.46_{\pm 7.62}$ | $85.88_{\pm 5.27}$ | $79.61_{\pm 6.85}$ | $77.55_{\pm 5.63}$ | $71.06_{\pm 7.71}$ |
| LDGCN-1% | $10.74_{\pm 2.87}$ | $9.25_{\pm 3.04}$ | $62.86_{\pm 4.54}$ | $57.82_{\pm 4.34}$ | $26.62_{\pm 4.76}$ | $23.15_{\pm 5.58}$ |
| D2GCN-10% | $70.99_{\pm 4.79}$ | $72.42_{\pm 4.74}$ | $83.24_{\pm 1.54}$ | $84.59_{\pm 2.71}$ | $74.08_{\pm 2.61}$ | $75.44_{\pm 4.11}$ |
| LDGCN-10% | $13.72_{\pm 2.62}$ | $12.90_{\pm 2.10}$ | $62.89_{\pm 2.43}$ | $59.82_{\pm 2.81}$ | $28.65_{\pm 2.71}$ | $26.23_{\pm 2.99}$ |

Table 5: Overlap Rate of D2GCN and LDGCN on Computers

| METHOD | $OVR_{node}$ | | $OVR_{cls}$ | | $OVR_{5\times cls}$ | |
|---|---|---|---|---|---|---|
| | 4-Layer | 6-Layer | 4-Layer | 6-Layer | 4-Layer | 6-Layer |
| D2GCN-1% | $99.7_{\pm 0.04}$ | $99.69_{\pm 0.05}$ | $99.84_{\pm 0.02}$ | $99.79_{\pm 0.04}$ | $99.77_{\pm 0.02}$ | $99.72_{\pm 0.05}$ |
| LDGCN-1% | $40.45_{\pm 1.64}$ | $43.27_{\pm 6.11}$ | $93.83_{\pm 0.98}$ | $94.23_{\pm 1.30}$ | $69.61_{\pm 1.36}$ | $68.99_{\pm 1.42}$ |
| D2GCN-10% | $99.71_{\pm 0.03}$ | $99.74_{\pm 0.03}$ | $99.83_{\pm 0.02}$ | $99.83_{\pm 0.01}$ | $99.74_{\pm 0.03}$ | $99.76_{\pm 0.03}$ |
| LDGCN-10% | $52.59_{\pm 0.66}$ | $54.85_{\pm 0.87}$ | $96.05_{\pm 1.39}$ | $94.61_{\pm 0.63}$ | $74.01_{\pm 1.14}$ | $74.29_{\pm 0.70}$ |

Table 6: Overlap Rate of D2GCN and LDGCN on CS

| METHOD | $OVR_{node}$ | | $OVR_{cls}$ | | $OVR_{5\times cls}$ | |
|---|---|---|---|---|---|---|
| | 4-Layer | 6-Layer | 4-Layer | 6-Layer | 4-Layer | 6-Layer |
| D2GCN-1% | $95.53_{\pm 0.95}$ | $95.18_{\pm 0.41}$ | $96.67_{\pm 0.75}$ | $96.38_{\pm 0.24}$ | $95.81_{\pm 0.89}$ | $95.46_{\pm 0.37}$ |
| LDGCN-1% | $24.73_{\pm 1.19}$ | $30.32_{\pm 3.02}$ | $59.76_{\pm 1.76}$ | $66.61_{\pm 1.20}$ | $34.57_{\pm 1.39}$ | $41.72_{\pm 0.38}$ |
| D2GCN-10% | $95.58_{\pm 0.30}$ | $95.37_{\pm 0.10}$ | $96.80_{\pm 0.25}$ | $96.59_{\pm 0.09}$ | $95.89_{\pm 0.31}$ | $95.68_{\pm 0.10}$ |
| LDGCN-10% | $25.48_{\pm 0.75}$ | $25.77_{\pm 0.14}$ | $63.23_{\pm 1.17}$ | $62.13_{\pm 0.10}$ | $36.33_{\pm 1.28}$ | $35.18_{\pm 0.36}$ |

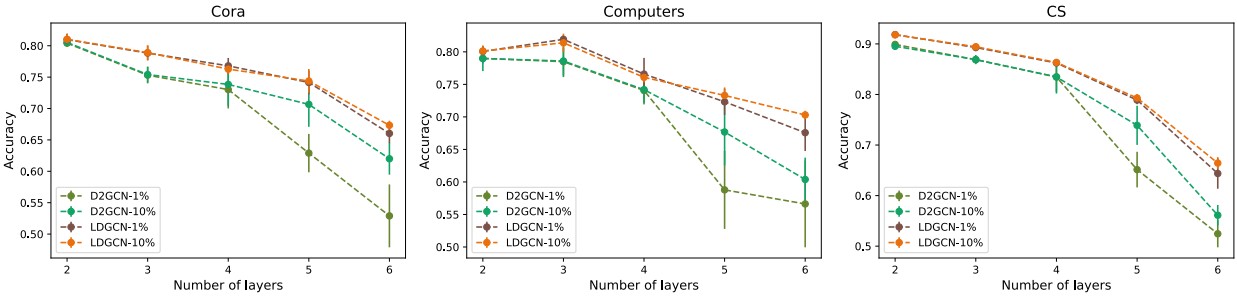

Figure 8: Compare the accuracy of LDGCN and D2GCN by choosing 1% and 10% nodes to perform negative sampling in three datasets.

## 4.2 Evaluation of Layer-diversity

This section presents a comparison of our LDGCN model with the previous D2GCN (Duan et al., 2022) to demonstrate that our approach effectively reduces the overlap rate of negative samples in terms of both nodes and clusters, and that these samples are more beneficial for graph learning.

**Datasets.** Three of seven previous datasets were chosen to test this claim: the citation network **Cora**, the coauthor network **CS**, and the Amazon network **Computers**. The average degrees of the three datasets are **3.90**, **8.93** and **35.76**, respectively. This difference in density facilitates the comparison of the two methods in different graph datasets.

**Setup.** We repeated the experiments using both 1% and 10% of the nodes selected for negative sampling. Our aim was to show that: (1) the layer-diverse projection method can identify a set of negative samples with reduced overlap between layers and less redundant information; (2) an efficient sampling method can achieve better performance with fewer central nodes.

**Metric.** In addition to utilizing accuracy to display the final prediction results, we developed two metrics to evaluate the overlap rate of the selected samples: the **Node Overlap Rate** ($\mathrm{OVR_{node}}$) and **Cluster Overlap Rate** ($\mathrm{OVR_{cls}}$). $\mathrm{OVR_{node}}$ assesses the average overlap of the samples in the last and current layers of the network, defined by

$$\mathrm{OVR_{node}} = \frac{1}{L}\frac{1}{|\mathbb{V}_c|}\sum_{l=2}^{L}\sum_{i\in\mathbb{V}_c}\frac{|\overline{\mathbb{N}}_i^{l-1}\cap\overline{\mathbb{N}}_i^{l}|}{|\overline{\mathbb{N}}_i^{l}|}, \tag{27}$$

where $\mathbb{V}_c$ are the central nodes performing the negative sampling, and $L$ is the number of network layers.

In addition to selecting diverse nodes, it is crucial that the selected nodes come from different clusters, as shown in Figure 1. In the context of semi-supervised learning with GNNs, we assume that the true labels of all nodes are currently unknown. Therefore, we employ K-Means to partition all nodes $\mathbb{V}$ into $K$ clusters $\mathbb{V} = \cup_{k=1}^{K}\mathbb{K}_k$ after each layer. This ensures that the negative samples $\overline{\mathbb{N}}_i^l$ for each layer belong to different clusters $\mathbb{K}_k$ and form the cluster set $\overline{\mathbb{C}}_i^l$. Then, we define $\mathrm{OVR_{cls}}$ to measure the overlap of the cluster sets between layers:

$$\mathrm{OVR_{cls}} = \frac{1}{L}\frac{1}{|\mathbb{V}_c|}\sum_{l=2}^{L}\sum_{i\in\mathbb{V}_c}\frac{\left|\overline{\mathbb{C}}_i^{l-1}\cap\overline{\mathbb{C}}_i^{l}\right|}{|\overline{\mathbb{C}}_i^{l}|}. \tag{28}$$

**Results.** Table 4, 5, and 6 illustrate the various overlap rates for the two methods on three datasets. To evaluate the cluster overlap rate, the number of clusters $K$ was set to: (a) the actual number of classes, denoted as $\mathrm{OVR_{cls}}$, and (b) 5 times the actual number of classes, denoted as $\mathrm{OVR_{5\times cls}}$. On all three datasets, we found that in comparison to D2GCN, LDGCN not only significantly decreased the node overlap rate but also reduced the repetition rate of the clusters to which the nodes belong. An interesting observation is that when we increased the number of clusters from the actual number of classes to 5 times the actual number of classes, LDGCN saw a further significant reduction in the sample overlap rate, while D2GCN only saw a slight reduction.

One possible explanation is that within each class, nodes can be further clustered. For instance, in the citation network, articles classified as machine learning can be further divided into articles on CNNs, GNNs, etc. Our layer-diverse method is designed to find the most diverse samples possible, even when searching within the same class, such as those belonging to these more specific clusters. In contrast, D2GCN consistently selects negative samples from the same cluster. These results demonstrate that LDGCN provides more useful information about the entire graph for feature extraction than D2GCN during message passing.

We evaluated the accuracy of LDGCN and D2GCN on the **Cora**, **Computers**, and **CS** datasets using different negative sampling rates. As shown in Figure 8, LDGCN demonstrated superior performance on all three datasets using both 1% and 10% of nodes for sampling. Examining the 1% experiment, where fewer nodes were used for negative message passing, we found that the selected nodes were meaningful enough to aid in graph learning. LDGCN maintained consistent performance even with a reduced sampling rate, while D2GCN's performance decreased significantly.

## 4.3 Evaluation of different GNNs architectures

This section presents an investigation of applying our layer-diverse sampling method to different GNN architectures on the **Cora**, **CS** and **Computers** datasets, which have different graph densities.

Table 7: Applying the layer-diverse sampling method to different GNN architectures on Cora

| Method | Layer 2 | Layer 4 | Layer 6 | Layer 8 | Layer 16 | Layer 32 |
|---|---|---|---|---|---|---|
| GCN | $80.03_{\pm0.52}$ | $63.39_{\pm7.92}$ | $17.16_{\pm3.24}$ | $13.90_{\pm0.55}$ | $14.19_{\pm9.56}$ | $14.33_{\pm0.79}$ |
| LDGCN | $\mathbf{80.94}_{\pm0.76}$ | $\mathbf{76.80}_{\pm1.26}$ | $\mathbf{67.91}_{\pm0.82}$ | $\mathbf{52.80}_{\pm5.96}$ | $\mathbf{30.12}_{\pm1.05}$ | $\mathbf{25.25}_{\pm1.08}$ |
| GATv2 | $78.36_{\pm1.66}$ | $74.43_{\pm3.80}$ | $62.03_{\pm6.60}$ | $30.75_{\pm2.24}$ | $23.17_{\pm8.18}$ | $22.72_{\pm5.60}$ |
| LDGATv2 | $\mathbf{79.30}_{\pm0.61}$ | $\mathbf{78.46}_{\pm0.85}$ | $\mathbf{67.79}_{\pm2.34}$ | $\mathbf{30.93}_{\pm0.00}$ | $\mathbf{25.11}_{\pm9.03}$ | $\mathbf{24.28}_{\pm8.30}$ |
| SAGE | $80.19_{\pm0.60}$ | $73.13_{\pm3.54}$ | $58.83_{\pm4.28}$ | $18.51_{\pm6.54}$ | $16.96_{\pm4.66}$ | $16.56_{\pm3.45}$ |
| LDSAGE | $\mathbf{80.26}_{\pm0.45}$ | $\mathbf{76.55}_{\pm0.94}$ | $\mathbf{65.39}_{\pm3.49}$ | $\mathbf{23.25}_{\pm0.16}$ | $\mathbf{20.89}_{\pm3.74}$ | $\mathbf{20.79}_{\pm4.51}$ |
| GIN | $78.95_{\pm1.02}$ | $68.07_{\pm4.26}$ | $36.36_{\pm6.80}$ | $29.36_{\pm3.80}$ | $27.47_{\pm3.60}$ | $15.86_{\pm8.00}$ |
| LDGIN | $\mathbf{79.21}_{\pm0.56}$ | $\mathbf{69.27}_{\pm1.70}$ | $\mathbf{39.34}_{\pm7.51}$ | $\mathbf{31.80}_{\pm3.33}$ | $\mathbf{31.79}_{\pm6.98}$ | $\mathbf{30.14}_{\pm7.13}$ |

Table 8: Applying the layer-diverse sampling method to different GNN architectures on CS

| Method | Layer 2 | Layer 4 | Layer 6 | Layer 8 | Layer 16 | Layer 32 |
|---|---|---|---|---|---|---|
| GCN | $90.71_{\pm0.91}$ | $54.00_{\pm3.69}$ | $40.77_{\pm3.52}$ | $24.92_{\pm12.28}$ | $10.26_{\pm3.37}$ | $9.02_{\pm2.00}$ |
| LDGCN | $\mathbf{91.53}_{\pm0.41}$ | $\mathbf{86.23}_{\pm0.55}$ | $\mathbf{64.37}_{\pm3.01}$ | $\mathbf{53.96}_{\pm0.87}$ | $\mathbf{19.82}_{\pm2.75}$ | $\mathbf{15.53}_{\pm1.15}$ |
| GATv2 | $89.28_{\pm1.62}$ | $85.00_{\pm1.55}$ | $52.19_{\pm7.86}$ | $34.24_{\pm12.67}$ | $21.89_{\pm3.03}$ | $22.90_{\pm0.00}$ |
| LDGATv2 | $\mathbf{90.26}_{\pm1.07}$ | $\mathbf{88.70}_{\pm1.24}$ | $\mathbf{75.59}_{\pm4.00}$ | $\mathbf{35.70}_{\pm7.01}$ | $\mathbf{22.90}_{\pm0.00}$ | $22.90_{\pm0.00}$ |
| SAGE | $90.64_{\pm0.63}$ | $82.22_{\pm2.60}$ | $59.60_{\pm7.16}$ | $22.14_{\pm10.26}$ | $10.26_{\pm3.37}$ | $9.20_{\pm2.00}$ |
| LDSAGE | $\mathbf{91.60}_{\pm0.31}$ | $\mathbf{86.33}_{\pm0.62}$ | $\mathbf{64.79}_{\pm1.58}$ | $\mathbf{47.54}_{\pm9.10}$ | $\mathbf{13.83}_{\pm3.66}$ | $\mathbf{12.80}_{\pm2.76}$ |
| GIN | $90.80_{\pm0.51}$ | $59.00_{\pm10.52}$ | $20.19_{\pm5.76}$ | $20.16_{\pm6.83}$ | $21.46_{\pm4.00}$ | $6.98_{\pm1.81}$ |
| LDGIN | $\mathbf{90.88}_{\pm0.72}$ | $\mathbf{66.10}_{\pm3.36}$ | $\mathbf{22.37}_{\pm4.25}$ | $\mathbf{20.61}_{\pm3.56}$ | $20.40_{\pm2.03}$ | $\mathbf{8.27}_{\pm3.31}$ |

Table 9: Applying the layer-diverse sampling method to different GNN architectures on Computers

| METHOD | LAYER 2 | LAYER 4 | LAYER 6 | LAYER 8 | LAYER 16 | LAYER 32 |
|---|---|---|---|---|---|---|
| GCN | $61.47_{\pm3.14}$ | $47.21_{\pm4.22}$ | $47.81_{\pm7.72}$ | $25.12_{\pm5.38}$ | $22.21_{\pm2.32}$ | $24.33_{\pm5.83}$ |
| LDGCN | $\mathbf{80.81}_{\pm0.26}$ | $\mathbf{77.92}_{\pm2.34}$ | $\mathbf{70.30}_{\pm0.65}$ | $\mathbf{59.39}_{\pm1.28}$ | $\mathbf{55.85}_{\pm2.00}$ | $\mathbf{54.50}_{\pm2.82}$ |
| GATv2 | $70.98_{\pm3.83}$ | $61.90_{\pm5.38}$ | $24.74_{\pm10.45}$ | $23.86_{\pm9.71}$ | $23.86_{\pm10.86}$ | $22.90_{\pm0.00}$ |
| LDGATv2 | $\mathbf{72.59}_{\pm1.02}$ | $\mathbf{70.28}_{\pm2.09}$ | $\mathbf{28.28}_{\pm20.47}$ | $\mathbf{31.10}_{\pm11.08}$ | $22.90_{\pm0.00}$ | $22.90_{\pm0.00}$ |
| SAGE | $80.75_{\pm0.84}$ | $59.27_{\pm7.85}$ | $21.59_{\pm9.67}$ | $20.89_{\pm9.91}$ | $19.89_{\pm8.86}$ | $19.23_{\pm8.09}$ |
| LDSAGE | $\mathbf{80.92}_{\pm0.41}$ | $\mathbf{75.61}_{\pm1.91}$ | $\mathbf{63.87}_{\pm1.23}$ | $\mathbf{56.06}_{\pm6.89}$ | $\mathbf{33.28}_{\pm0.67}$ | $\mathbf{37.29}_{\pm7.52}$ |
| GIN | $33.73_{\pm6.25}$ | $37.09_{\pm2.21}$ | $35.80_{\pm0.39}$ | $35.65_{\pm0.20}$ | $6.62_{\pm5.84}$ | $3.06_{\pm0.00}$ |
| LDGIN | $\mathbf{35.38}_{\pm2.12}$ | $36.67_{\pm1.30}$ | $\mathbf{39.34}_{\pm7.51}$ | $\mathbf{36.76}_{\pm4.22}$ | $\mathbf{7.39}_{\pm5.32}$ | $3.06_{\pm0.00}$ |

**Setup.** Besides GCN (Kipf & Welling, 2017), layer-diverse sampling method was applied to GATv2 (Brody et al., 2022), SAGE (Hamilton et al., 2017) and GIN-$\epsilon$ (Xu et al., 2019), which were called LDGATv2, LDSAGE, LDGIN-$\epsilon$ seperately in the following. We repeated the experiments using 1% of the nodes selected for negative sampling. The layers of different models are set as $\{2, 4, 6, 8, 16, 32\}$. Our aim was to show that: the layer-diverse negative sampling is applicable to different GNN architectures and helps theses models to relieve the over-smoothing problem so that to achieve better results when the layers of the model become deeper.

**Results.** The results are shown Table.7, 8 and 9, which compare the performance of GNN architectures with and without layer-diverse negative sampling methods on the Cora and CS datasets. It's clear that layer-diverse methods (LDGCN, LDGATv2, LDSAGE, LDGIN) consistently outperform their counterparts

Table 10: MAD of 4-layer NegGCN models on all datasets

|  | Citeseer | Cora | PubMed | Coauthor-CS | Computers | Photo | ogbn-arxiv |
|---|---|---|---|---|---|---|---|
| GCN | $66.46_{\pm6.35}$ | $70.97_{\pm5.72}$ | $76.97_{\pm5.72}$ | $63.76_{\pm5.19}$ | $50.08_{\pm4.88}$ | $60.78_{\pm5.66}$ | $9.78_{\pm0.11}$ |
| RGCN | $74.08_{\pm6.07}$ | $75.22_{\pm4.44}$ | $87.18_{\pm7.61}$ | $73.13_{\pm3.91}$ | $55.70_{\pm6.88}$ | $73.07_{\pm6.68}$ | $77.17_{\pm2.49}$ |
| MCGCN | $70.49_{\pm7.69}$ | $74.40_{\pm5.51}$ | $80.52_{\pm4.41}$ | $72.41_{\pm3.75}$ | $55.56_{\pm6.04}$ | $71.20_{\pm4.97}$ | $76.32_{\pm0.67}$ |
| PGCN | $70.89_{\pm7.31}$ | $74.86_{\pm5.07}$ | $85.33_{\pm9.37}$ | $73.24_{\pm3.15}$ | $57.81_{\pm6.34}$ | $74.75_{\pm6.40}$ | $75.91_{\pm1.05}$ |
| D2GCN | $73.93_{\pm6.22}$ | $73.15_{\pm5.06}$ | $83.20_{\pm8.15}$ | $72.51_{\pm3.02}$ | $57.34_{\pm4.50}$ | $75.90_{\pm5.65}$ | $80.47_{\pm0/60}$ |
| LDGCN | $\mathbf{74.71}_{\pm2.60}$ | $\mathbf{75.24}_{\pm3.69}$ | $\mathbf{88.88}_{\pm7.94}$ | $\mathbf{73.25}_{\pm3.57}$ | $\mathbf{62.33}_{\pm6.69}$ | $\mathbf{79.45}_{\pm4.19}$ | $\mathbf{81.09}_{\pm0.91}$ |

Table 11: MAD of 6-layer NegGCN models on all datasets

|  | Citeseer | Cora | PubMed | Coauthor-CS | Computers | Photo | ogbn-arxiv |
|---|---|---|---|---|---|---|---|
| GCN | $7.44_{\pm5.04}$ | $6.68_{\pm3.46}$ | $75.94_{\pm7.26}$ | $62.57_{\pm3.62}$ | $46.16_{\pm6.74}$ | $57.88_{\pm6.06}$ | $8.96_{\pm0.20}$ |
| RGCN | $45.98_{\pm19.98}$ | $64.71_{\pm6.07}$ | $76.52_{\pm4.57}$ | $69.73_{\pm5.12}$ | $54.74_{\pm8.24}$ | $79.16_{\pm4.19}$ | $76.04_{\pm1.35}$ |
| MCGCN | $57.33_{\pm16.78}$ | $67.60_{\pm5.26}$ | $75.67_{\pm6.70}$ | $67.82_{\pm1.40}$ | $56.67_{\pm4.44}$ | $77.88_{\pm4.49}$ | $72.81_{\pm0.75}$ |
| PGCN | $50.95_{\pm18.73}$ | $69.02_{\pm5.95}$ | $76.03_{\pm3.89}$ | $71.16_{\pm3.95}$ | $57.35_{\pm1.65}$ | $76.76_{\pm5.11}$ | $73.01_{\pm1.14}$ |
| D2GCN | $71.79_{\pm2.39}$ | $70.51_{\pm5.59}$ | $77.57_{\pm6.93}$ | $72.22_{\pm2.07}$ | $57.45_{\pm6.64}$ | $78.83_{\pm7.82}$ | $77.87_{\pm0.51}$ |
| LDGCN | $\mathbf{74.29}_{\pm2.27}$ | $\mathbf{74.92}_{\pm5.56}$ | $\mathbf{81.40}_{\pm3.55}$ | $\mathbf{73.70}_{\pm1.39}$ | $\mathbf{62.18}_{\pm5.11}$ | $\mathbf{82.67}_{\pm1.43}$ | $\mathbf{78.33}_{\pm1.24}$ |

without layer-diverse methods (GCN, GATv2, SAGE, GIN) across all layer sizes for both datasets. Although as the layers in the network increased, all models showed a trend of decreasing accuracy, layer-diverse methods consistently performed better, implying that these methods enhance the effectiveness of GNNs in capturing informative samples for learning better graph representations. For example, LDGATv2 outperforms GATv2 on CS, with the highest performance improvement observed in Layer 6 (from 52.19 to 75.59); LDSAGE outperforms SAGE in all layer sizes on Computers, with the highest performance improvement observed in Layer 6 (from 21.59 to 63.87).

Interestingly, it was observed that GIN failed to converge for all layer settings on the Computers dataset. As a result, the layer-diverse method, which had consistently shown improvements on other GNN architectures, couldn't enhance the performance of GIN on the Computers dataset. This outcome underlines the fact that the layer-diverse approach may not be universally applicable or beneficial for all GNN architectures and datasets and that individual characteristics of the networks and the data can play a significant role.

In summary, the layer-diverse negative sampling methods have consistently improved performance across various architectures and datasets, supporting their effectiveness in graph-based learning tasks. They have potential for further exploration in other tasks or architectures, and could be a promising direction for improving the performance of GNNs, especially those with multiple layers.

## 4.4 Evaluation of Over-smoothing and Over-squashing

**Over-smoothing.** To measure the smoothness of the graph representations, we employed the Mean Average Distance (MAD) metric (Chen et al., 2020), which was computed as $\text{MAD} = \frac{\sum_i D_i}{\sum_i 1(D_i)}$, where $D_i = \frac{\sum_j \boldsymbol{D}_{ij}}{\sum_j 1(\boldsymbol{D}_{ij})}$, and $\boldsymbol{D}_{ij} = 1 - \cos(x_i, x_j)$ is the cosine distance between the nodes $i$ and $j$. Our comparison between LDGCN and other negative sampling methods are presented in Table 10 and Table 11. As can be seen from these results, LDGCN's MAD is higher than the other methods on all datasets. All the negative sampling methods, except for GCN, had relatively high MADs, indicating that adding negative samples to the message passing increases the distance between nodes. These results confirm our argument

Table 12: Assessing over-squashing using accuracy and MAD on Cora-based graph

|  | ACC | MAD |
|---|---|---|
| GCN | $65.24_{\pm 6.19}$ | $57.39_{\pm 0.00}$ |
| GCN+FA | $43.64_{\pm 0.00}$ | $0.00_{\pm 0.00}$ |
| LDGCN | $71.17_{\pm 5.03}$ | $74.48_{\pm 2.19}$ |

Table 13: Computational time per epoch and per run for various methods on Citeseer

| Methods | Time (s) /Epoch | Time (s) /Run |
|---|---|---|
| GCN | $0.01 \pm 0.00$ | $1.58 \pm 0.15$ |
| SAGE | $0.01 \pm 0.00$ | $1.20 \pm 0.11$ |
| GATv2 | $0.01 \pm 0.00$ | $2.18 \pm 1.81$ |
| GIN | $0.01 \pm 0.00$ | $1.16 \pm 0.18$ |
| AERO | $0.07 \pm 0.00$ | $14.53 \pm 3.46$ |
| MCGCN | $0.39 \pm 0.01$ | $41.76 \pm 0.29$ |
| PGCN | $0.01 \pm 0.00$ | $1.17 \pm 0.87$ |
| RGCN | $0.01 \pm 0.00$ | $1.98 \pm 1.83$ |
| D2GCN-1% | $0.25 \pm 0.05$ | $47.73 \pm 2.62$ |
| LDGCN-1% | $0.21 \pm 0.04$ | $39.30 \pm 3.38$ |
| D2GCN-10% | $2.10 \pm 0.11$ | $431.89 \pm 13.03$ |
| LDGCN-10% | $1.73 \pm 0.08$ | $346.01 \pm 13.01$ |

in Section 3.2 that incorporating negative samples into the convolution increases the upper bound of the distance between nodes.

**Over-squashing.** Using the Cora dataset, we created a graph $\mathcal{G}_o$ with a bottleneck where only one edge linked two distinct communities. We then compared LDGCN with the basic GCN method (Kipf & Welling, 2017) and the GCN+FA method (Alon & Yahav, 2021), where the last layer of GCN was fully connected. More information on the methods used to construct the graph, as well as the experiment settings, can be found in Appendix A.3.3. Table 12 presents the results in terms of accuracy and MAD. GCN+FA added too many edges in the graph at the last layer, which resulted in over-smoothing problems, and MAD went to zero. In contrast, our method reduces the likelihood of over-squashing and improves classification accuracy without the negative side effect of over-smoothing.

### 4.5 Evaluation of Time Complexity

The computational complexities of Eqs. (9) and (10) are $\mathcal{O}(S)$ and $\mathcal{O}(S^2)$ respectively for $S = |\mathbb{S}_i|$. Let $D$ be the average node degree, which is a constant for a given dataset and $D \ll S \ll N$. The complexity of the loop in Algorithm 2 is then $\mathcal{O}(DS^2)$. Since matrix decomposition is an essential step in DPP, with complexity $\mathcal{O}(S^3)$, if we take into account every node in the sampling process, the total one-time cost of our method would be $\mathcal{O}(N(DS^2 + S^3))$. To reduce this cost, we can do negative sampling only on a fractional number of nodes. Experiments in Section 4.2 show that negative sampling on (random) 1% or 10% nodes suffices to achieve good performance in general. Taking the Citeseer as an example, Tabel.13 shows the computational time per epoch and per run for various methods under 4 layers.

### 4.6 Case Study for Different Negative Sampling Methods

We conducted a case study using the Cora dataset to provide a qualitative comparison between our proposed Layer-Diverse Graph Convolutional Network (LDGCN) and the existing Determinantal Point Process (DPP) based model (D2GCN).

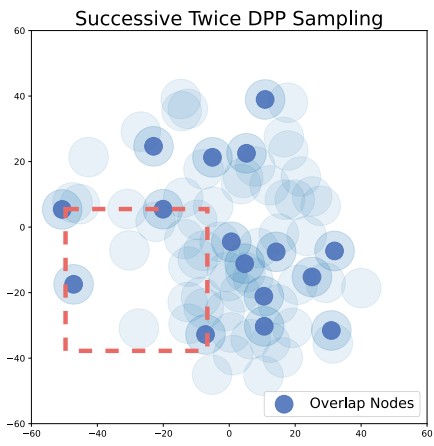 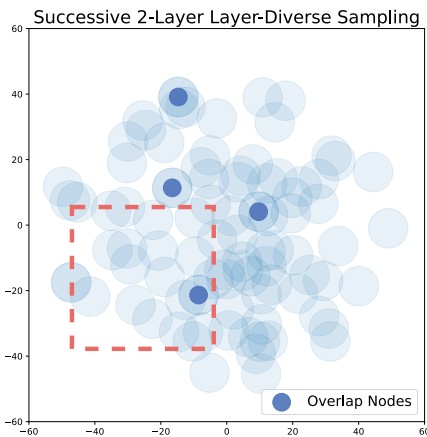

Figure 9: **Left:** Sampling results from the D2GCN model without layer diversity. **Right:** Sampling results from LDGCN model with layer-diverse. Overlap nodes, indicating sampling redundancy, are highlighted in dark blue. Sample diversity is shown in red dash box.

We implemented a 2-layer GNN for both the D2GCN (without layer diversity) and our LDGCN (with layer diversity). We employed the respective sampling strategies for each model and collected the indices of nodes sampled in two consecutive layers. When visualizing these nodes in a 2D space, we used the original node features as shown in Figure 9. The left panel depicts the sampling results from the D2GCN model, and the right panel illustrates the sampling by our LDGCN model. Overlap nodes from two successive layers are highlighted in dark blue for clarity.

From this case study, we observe two key outcomes:

- **Reduced overlap in sampling**: The LDGCN model demonstrates fewer overlap nodes than the D2GCN model. This finding substantiates our claim that layer-diverse negative sampling effectively reduces the likelihood of resampling duplicate nodes across layers.

- **Enhanced sample diversity**: Aside from reducing overlap nodes, the LDGCN model's samples are more uniformly distributed in the 2D space. This contrasts with the D2GCN model, where samples appear more clustered and thus exhibit a higher spatial overlap. The more dispersed samples from the LDGCN model suggest that layer diversity contributes to increased sample diversity and a more distinct representation for each layer.

## 5    Related Work

**GNNs and their variants.** GNNs are typically divided into two categories: spectral-based and spatial-based. A widely used and straightforward example of the first category is the graph convolutional network (GCN) (Kipf & Welling, 2017), which utilizes first-order approximations of spectral graph convolutions. Many variations have been proposed based on GCN, such as GPRGNN (Chien et al., 2021), GNN-LF/HF (Zhu et al., 2021), UFG (Zheng et al., 2021) which uses graph framelet transforms to define convolution. In the spatial stream, GraphSAGE (Hamilton et al., 2017) is a well-known model. It utilizes node attribute information to effectively generate representations for previously unseen data. Xu et al. (2019) provides a theoretical analysis of GNNs' representational power in capturing various graph structures and proposed the Graph Isomorphism Network. Besides these two models, there are numerous spatial-based methods, such as GAT (Velickovic et al., 2018) and PPNP (Klicpera et al., 2019) to mention just a few.

**Negative sampling in GNNs.** All the GNNs previously mentioned are based on positive sampling. In terms of negative sampling, there are roughly two kinds negative sampling methods for graph representation learning. The first includes methods such as randomly selecting (Kim & Oh, 2021), Monte Carlo chains based (Yang et al., 2020), and personalized Page-Rank based (Ying et al., 2018). While these methods do find negative samples, they often have a high degree of redundancy or the small clusters are overwhelmed by large clusters. These do not meet the criteria for obtaining good negative samples as proposed in (Duan et al., 2022). Duan et al. (2022) attempt to find negative samples that meet the above criteria, and focus on controlling the diversity of negative samples using DPP (Kulesza & Taskar, 2012). However, the found samples were still highly redundant, and it has not yet been confirmed whether these samples meet the criteria for being good negative samples.

**DPP and its applications.** Determinantal point process (DPP) was first introduced to the field of machine learning by Kulesza & Taskar (2012) as $k$-determinantal point process ($k$-DPP). The $k$-DPP is a generalization of the DPP for sampling a fixed number of items, $k$, rather than a variable number, which is defined by a positive semidefinite kernel matrix, and encodes the similarity between the items in the candidate set. The $k$-DPP method for negative sampling in graph representation learning is a way of selecting negative samples by controlling the diversity of negative samples using the $k$-DPP. This method is particularly effective in capturing the properties of repulsion and has been successfully applied to various scenarios such as sequential labelling (Qiao et al., 2015), document summarization (Cho et al., 2019), video summarization (Zheng & Lu, 2020).

## 6 Conclusion

In this paper, we presented a novel approach for negative sampling in graph representation learning, based on a layer-diverse DPP sampling method and space squeezing technique. Our method is able to significantly reduce the redundancy associated with negative sampling, resulting in improved overall classification performance. We also provided an in-depth analysis of why negative samples are beneficial for GNNs and how they help to address common issues such as over-smoothing, GNN expressivity, and over-squashing. Through extensive experiments, we confirmed that our method can effectively improve graph learning ability. Furthermore, our approach can be applied to various types of graph learning tasks, and it is expected to have a wide range of potential applications.

**Acknowledgments**

This work is supported by the Australian Research Council under Australian Laureate Fellowships FL190100149 and Discovery Early Career Researcher Award DE200100245.

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

---

**Algorithm 3:** Obtain a sample from $k$-DPP (Kulesza & Taskar, 2012)

---

**Input:** Ground set $\mathbb{Y}$, size $k$, matrix $\boldsymbol{L}$ used for sampling

**1** Eigen-decompose $\boldsymbol{L}$ to obtain eigenvalues $\lambda_1, \ldots, \lambda_{|\mathbb{Y}|}$;

**2** Iteratively calculate all elementary symmetric polynomials $e_l^v$ for $l = 0, \ldots, k$ and $v = 0, \ldots, |\mathbb{Y}|$
   following Eq. (33);

**3** $J \leftarrow \emptyset$;

**4** $l \leftarrow k$;

**5 for** $v = |\mathbb{Y}|, \ldots, 1$ **do**

**6**  | **if** $l = 0$ **then**

**7**  |  | break

**8**  | **end**

**9**  | **if** $\mu \sim U[0,1] < \lambda_v \frac{e_{l-1}^{v-1}}{e_l^v}$ **then**

**10** |  | $J \leftarrow J \cup \{v\}$;

**11** |  | $l \leftarrow l - 1$;

**12** | **end**

**13 end**

**14** $\mathbb{V} \leftarrow \{\boldsymbol{v}_n\}_{n \in J}$;

**15** $\mathbb{Y}_{\text{sub}} \leftarrow \emptyset$;

**16 while** $|\mathbb{V}| > 0$ **do**

**17** | Select $i$ from $\mathbb{Y}$ with $\Pr(i) = \frac{1}{|\mathbb{V}|} \sum_{\boldsymbol{v} \in V} \left(\boldsymbol{v}^\top \boldsymbol{e}_i\right)^2$;

**18** | $\mathbb{Y}_{\text{sub}} \leftarrow \mathbb{Y}_{\text{sub}} \cup i$;

**19** | $\mathbb{V} \leftarrow \mathbb{V}_\perp$, an orthonormal basis for the subspace of $\mathbb{V}$ orthogonal to $\boldsymbol{e}_i$;

**20 end**

**Output:** $\mathbb{Y}_{\text{sub}}$ with size $|\mathbb{Y}_{\text{sub}}| = k$

---

Meiqi Zhu, Xiao Wang, Chuan Shi, Houye Ji, and Peng Cui. Interpreting and unifying graph neural networks with an optimization framework. In Jure Leskovec, Marko Grobelnik, Marc Najork, Jie Tang, and Leila Zia (eds.), *WWW '21: The Web Conference (WWW 2021), Virtual Event / Ljubljana, Slovenia, April 19-23, 2021*, pp. 1215–1226, 2021.

## A  Appendix

### A.1  Detail for Determinantal Point Processes (DPP)

A point process on a ground set $\mathbb{Y}$ is a probability measure over "point patterns", which are finite subsets of $\mathbb{Y}$ (Kulesza & Taskar, 2012). A determinantal point process $\Psi$ is a probability measure on all possible subsets of the ground set $\mathbb{Y}$ with a size of $2^{|\mathbb{Y}|}$. For every subset $\mathbb{Y}_{\text{sub}} \subseteq \mathbb{Y}$, a DPP (Hough et al., 2009) defined via a positive semidefinite $\boldsymbol{L}$ matrix is formulated as

$$\Psi_{\boldsymbol{L}}(\mathbb{Y}_{\text{sub}}) = \frac{\det\left(\boldsymbol{L}_{\mathbb{Y}_{\text{sub}}}\right)}{\det(\boldsymbol{L} + \boldsymbol{I})}, \tag{29}$$

where $\det(\cdot)$ denotes the determinant of a given matrix, $\boldsymbol{L}$ is a real and symmetric $|\mathbb{Y}| \times |\mathbb{Y}|$ matrix indexed by the elements of $\mathbb{Y}$, and $\det(\boldsymbol{L} + \boldsymbol{I})$ is a normalisation term that is constant once the ground dataset $\mathbb{Y}$ is fixed.

DPP has an intuitive geometric interpretation. If we have a $\boldsymbol{L}$, there is always a matrix $\boldsymbol{B}$ that satisfies $\boldsymbol{L} = \boldsymbol{B}^\top \boldsymbol{B}$. Let $\boldsymbol{B}_i$ be the columns of $\boldsymbol{B}$. A determinantal operator can then be interpreted geometrically as

$$\Psi_{\boldsymbol{L}}(\mathbb{Y}_{\text{sub}}) \propto \det\left(\boldsymbol{L}_{\mathbb{Y}_{\text{sub}}}\right) = \text{vol}^2\left(\{\boldsymbol{B}_i\}_{i \in \mathbb{Y}_{\text{sub}}}\right), \tag{30}$$

where the right-hand side of the equation is the squared $|\mathbb{Y}_{\text{sub}}|$-dimensional volume of the parallelepiped spanned by the columns of $\boldsymbol{B}$ corresponding to the elements in $\mathbb{Y}_{\text{sub}}$. Intuitively, diverse sets are more probable because their feature vectors are more orthogonal and span larger volumes.

One important variant of DPP is $k$-DPP (Kulesza & Taskar, 2012). $k$-DPP measures only $k$-sized subsets of $\mathbb{Y}$ rather than all of them including an empty subset. It is formally defined as

$$\Psi_{\boldsymbol{L}}^k(\mathbb{Y}_{\text{sub}}) = \frac{\det\left(\boldsymbol{L}_{\mathbb{Y}_{\text{sub}}}\right)}{\sum_{|\mathbb{Y}'_{\text{sub}}|=k} \det\left(\boldsymbol{L}_{\mathbb{Y}'_{\text{sub}}}\right)}, \tag{31}$$

with the cardinality of subset $\mathbb{Y}_{\text{sub}}$ being a fixed size $k$, i.e., $|\mathbb{Y}_{\text{sub}}| = k$. Here, we use a sampling method based on eigendecomposition (Hough et al., 2006; Kulesza & Taskar, 2012). Eq. (31) can be rewritten as:

$$\Psi_{\boldsymbol{L}}^k(\mathbb{Y}_{\text{sub}}) = \frac{1}{e_k^{|\mathbb{Y}|}} \det(\boldsymbol{L} + \boldsymbol{I})\Psi_{\boldsymbol{L}}(\mathbb{Y}_{\text{sub}}), \tag{32}$$

where $e_k^{|\mathbb{Y}|}$ is the $k^{th}$ elementary symmetric polynomial on eigenvalues $\lambda_1, \lambda_2, \ldots, \lambda_{|\mathbb{Y}|}$ of $\boldsymbol{L}$ defined as:

$$e_k^{|\mathbb{Y}|} = e_k\left(\lambda_1, \lambda_2, \ldots, \lambda_{|\mathbb{Y}|}\right) = \sum_{\substack{J \subseteq \{1,2,\ldots,|\mathbb{Y}|\} \\ |J|=k}} \prod_{n \in J} \lambda_n. \tag{33}$$

Following Kulesza & Taskar (2012), Eq. (32) is decomposed into elementary parts as

$$\Psi_{\boldsymbol{L}}^k(\mathbb{Y}_{\text{sub}}) = \frac{1}{e_k^{|\mathbb{Y}|}} \sum_{|J|=k} \mathcal{P}^{V_J}(\mathbb{Y}_{\text{sub}}) \prod_{m \in J} \lambda_m, \tag{34}$$

where $V_J$ denotes the set $\{\boldsymbol{v}_m\}_{m \in J}$ and $\boldsymbol{v}_m$ and $\lambda_m$ are the eigenvectors and eigenvalues of the $\boldsymbol{L}$-ensemble, respectively. Based on Eq. (34), for the self-contained reason, the complete process of sampling from $k$-DPP is given in Algorithm 3. More details about DPP and $k$-DPP can be found in Kulesza & Taskar (2012).

Since the sampling procedure in the DPP requires an eigendecomposition, the computational cost for Algorithm 3 could be $O(|N|^3)$, where $N = |\mathbb{Y}|$. If the sampling is performed for every node in the graph, the total will become an excessive $O(|N|^4)$ for all nodes. Thus, the large size of candidates found from exploring the whole graph to find negative samples would make such an approach impractical, even for a moderately sized graph.

To reduce the computational complexity, the shortest-path-based method (Duan et al., 2023) is first used to form a smaller but more effective candidate set $\mathbb{S}_i$ for node $i$, which is detailed in Algorithm 1. Using this method, the computational cost is approximately $O((P \cdot \overline{\deg})^3)$, where $P \ll N$ is the path length (normally smaller than the diameter of the graph) and $\overline{\deg} \ll N$ is the average degree of the graph. As an example, consider a Citeseer graph (Sen et al., 2008) with 3,327 nodes and $\overline{\deg} = 2.74$. When using the experimental setting shown below, where $P = 5$, we observe that $O(N^3) = 3.6 \times 10^{10}$, which is significantly larger than $2571 = O((P \cdot \overline{\deg})^3)$.

## A.2 Remark of Space Squeeze

**Remark A.1.** *Suppose the probability of re-picking node $\bar{j}^* \in \overline{\mathbb{N}}_i^l$ in $\boldsymbol{V}$ is p, the new probability of re-picking it in $\boldsymbol{V}'$ would be reduced to $(1-\gamma)p$, where $0 \le \gamma \le 1$. It means that we can control the squeezing degree by $\gamma$.*

*Proof.* After the space squeezing using Eq.(10), the $\bar{j}^*$ row of new space is

$$
\begin{aligned}
\boldsymbol{V}'[\bar{j}^*,:] &= \boldsymbol{V}[\bar{j}^*,:] - \gamma\boldsymbol{V}[\bar{j}^*,m] \cdot \frac{\boldsymbol{V}[\bar{j}^*,:]}{\boldsymbol{V}[\bar{j}^*,m]} \\
&= \boldsymbol{V}[\bar{j}^*,:] - \gamma\boldsymbol{V}[\bar{j}^*,:] \\
&= (1-\gamma)\boldsymbol{V}[\bar{j}^*,:].
\end{aligned}
\tag{35}
$$

Since the probability of picking node $\bar{j}^*$ through the DPP sampling is proportional to $||\boldsymbol{V}[\bar{j}^*,:]||_2$, we denote the probability of re-picking node $\bar{j}^* \in \overline{\mathbb{N}}_i^l$ in $\boldsymbol{V}$ is

$$
p = ||\boldsymbol{V}[\bar{j}^*,:]||_2.
\tag{36}
$$

According to Eqs. (35) and (36), the new probability of re-picking it in $\boldsymbol{V}'$ is $(1-\gamma)p$. $\qquad\square$

**Remark A.2.** *For a node $\bar{i} \in \overline{\mathbb{N}}_i^l$ and $\bar{i} \neq \bar{j}^*$, if $\boldsymbol{V}[\bar{i},:]$ and $\boldsymbol{V}[\bar{j}^*,:]$ are sufficiently similar with each other, then the probability of re-picking $\bar{i}$ would also be reduced. It means that we do not just reduce the re-picking probability of $\bar{j}^*$. By reducing the re-picking probability of $\bar{j}^*$, we also decrease the influence of similar nodes, reducing the likelihood of them being considered.*

*Proof.* For any two $L$-length vectors $\boldsymbol{v}_1$ and $\boldsymbol{v}_2$, if the following conditions are satisfied,

$$
\boldsymbol{\varphi} = \frac{\boldsymbol{v}_1}{\boldsymbol{v}_2}, \quad 1-\delta \leq \boldsymbol{\varphi}[m] \leq 1+\delta \quad \text{for any } 0 \leq m \leq L,
\tag{37}
$$

then we call $\boldsymbol{v}_1$ and $\boldsymbol{v}_2$ are $\delta$-similar with each other.

For a node $\bar{i} \in \overline{\mathbb{N}}_i^l$ and $\bar{i} \neq \bar{j}^*$, the new representation of $\bar{i}$ in new space $\boldsymbol{V}'$ would be

$$
\boldsymbol{V}'[\bar{i},:] = \boldsymbol{V}[\bar{i},:] - \gamma\boldsymbol{V}[\bar{i},m] \cdot \frac{\boldsymbol{V}[\bar{j}^*,:]}{\boldsymbol{V}[\bar{j}^*,m]}.
\tag{38}
$$

If $\boldsymbol{V}[\bar{i},:]$ and $\boldsymbol{V}[\bar{j}^*,:]$ are $\delta$-similar with each other, we have

$$
\boldsymbol{V}'[\bar{i},:] = \boldsymbol{\varphi} \odot \boldsymbol{V}[\bar{i},:] - \gamma\boldsymbol{V}[\bar{i},m] \cdot \frac{\boldsymbol{V}[\bar{j}^*,:]}{\boldsymbol{V}[\bar{j}^*,m]},
\tag{39}
$$

and according to conditions in (37), we have

$$
((1-\delta) - \gamma(1+\delta))\boldsymbol{V}[\bar{j}^*,:] \leq \boldsymbol{V}'[\bar{i},:] \leq ((1+\delta) - \gamma(1-\delta))\boldsymbol{V}[\bar{j}^*,:].
\tag{40}
$$

When $\delta$ goes to 0, according to sandwich theorem, we have

$$
\lim_{\delta \to 0} \boldsymbol{V}[\bar{i},:] = (1-\gamma)\boldsymbol{V}[\bar{j}^*,:].
\tag{41}
$$

That is to say, if node $\bar{i}$ is similar enough with $\bar{j}^*$, the space squeezing will also reduce the probability of selecting node $\bar{i}$. The more similar the two nodes are, the lower node $\bar{i}$ will be re-picked. Hence, we do not just reduce the re-picking probability of $\bar{j}^*$ but also the information of this node, and any nodes sharing similar information with this node would not be considered with high probability. $\qquad\square$

### A.3 Experiment Details

#### A.3.1 Dataset statistics

The datasets are split generally following Kipf & Welling (2017). For the first 6 datasets, we choose 20 nodes for each class as the training set. For the Ogbn-arxiv, because this graph is large, we choose 100 nodes for each class as the training set.

The first six datasets are downloaded from PyTorch Geometric (PyG)[2]. The Ogbn-arxiv is downloaded from Open Graph Benchmark (OGB)[3].

---

[2]https://pytorch-geometric.readthedocs.io/en/latest/modules/datasets.html
[3]https://ogb.stanford.edu/docs/nodeprop/

Table 14: Dataset Statisric

| Dataset | Nodes | Edges | Classes | Features | Average of Degree | Label Rate | Val / Test | Epoch |
|---|---|---|---|---|---|---|---|---|
| Citeseer | 3,327 | 9,104 | 6 | 3,703 | 2.74 | 3.61 % | 500/1000 | 200 |
| Cora | 2,708 | 10,556 | 7 | 1,443 | 3.90 | 5.17 % | 500/1000 | 200 |
| PubMed | 19,717 | 88,648 | 3 | 500 | 4.50 | 0.30 % | 500/1000 | 200 |
| CoauthorCS | 18,333 | 163,788 | 15 | 6805 | 8.93 | 1.64% | 500/1000 | 200 |
| Computers | 13,752 | 491,722 | 10 | 767 | 35.76 | 1.45% | 500/1000 | 200 |
| Photo | 7,650 | 238,162 | 8 | 745 | 31.13 | 2.09% | 500/1000 | 400 |
| Ogbn-arxiv | 169,343 | 1,166,243 | 40 | 128 | 35.76 | 53.70% | 29799/48603 | 200 |
| Cornell | 183 | 298 | 5 | 1703 | 1.63 | 48% | 59/37 | 200 |
| Texas | 183 | 325 | 5 | 1703 | 1.78 | 48% | 59/37 | 200 |
| Wisconsin | 251 | 515 | 5 | 1703 | 2.05 | 48% | 80/51 | 200 |

### A.3.2 Implementation Details

The experimental task was standard node classification. We set the maximum length of the shortest path $P$ to 6 in Algorithm 1, which means after throwing away the first-order nearest neighbours, there are still 5 nodes at the end of the different shortest paths. The size of the candidate set for node i is $|\mathbb{S}_i| = 5 \times D$. When selecting 1% or 10% nodes to perform negative sampling, we choose nodes whose degree is greater than the average degree $D$.

The negative rate $\mu$ is a trainable parameter and is trained in all models. Each model was trained using an Adam optimiser with a learning rate of 0.02. The number of hidden channels is set to 16 for all models. Tests for each model with each dataset were conducted ten times. The convolution layers of GCN (Kipf & Welling, 2017), SAGE (Hamilton et al., 2017), GATv2 (Brody et al., 2022) and GIN-$\epsilon$ (Xu et al., 2019) use PyTorch Geometric to implement [4]. The AERO (Lee et al., 2023) is implemented by the code published on Github [5]. All experiments were conducted on an Intel(R) Xeon(R) Gold 6326 CPU @ 2.90GHz and NVIDIA A100 PCIe 80GB GPU. The software that we use for experiments is Python 3.7.13, PyTorch 1.12.1, torch-geometric 2.1.0, torch-scatter 2.0.9, torch-sparse 0.6.15, torchvision 0.13.1, ogb 1.3.4, numpy 1.21.5 and CUDA 11.6.

---

**Algorithm 4:** Constructing the graph $\mathcal{G}_o$ having bottlenecks

**Input** : Number of communities $Q$, original graph $\mathcal{G}$

**1** Using Fluid Communities method (Parés et al., 2017) to get $Q$ communities in $\mathcal{G}$;
**2 while** *True* **do**
**3**     Delete the node linking two different communities;
**4**     **if** *only one node linking two different communities* **then**
**5**        **Break**;
**6**     **end**
**7 end**
**8** Obtain the maximum connected subgraph from the remaining graph as $\mathcal{G}_o$;
**Output:** Graph $\mathcal{G}_o$

---

### A.3.3 Over-squashing Experiment Details

**Algorithm of constructing the graph having bottlenecks.** The central concept in construction is to segment the initial graph into distinct clusters and incrementally eliminate the nodes linking these clusters

---

[4]https://pytorch-geometric.readthedocs.io/en/latest/modules/nn.html#convolutional-layers
[5]https://github.com/syleeheal/AERO-GNN/

Table 15: STATISTICS of $\mathcal{G}_o$

| Dataset | Nodes | Edges | Classes | Features | Average of Degree | Label Rate | Val / Test | Epoch |
|---|---|---|---|---|---|---|---|---|
| $\mathcal{G}_o$ (Cora-based) | 915 | 3054 | 7 | 1443 | 3.33 | 8% | 400/400 | 200 |

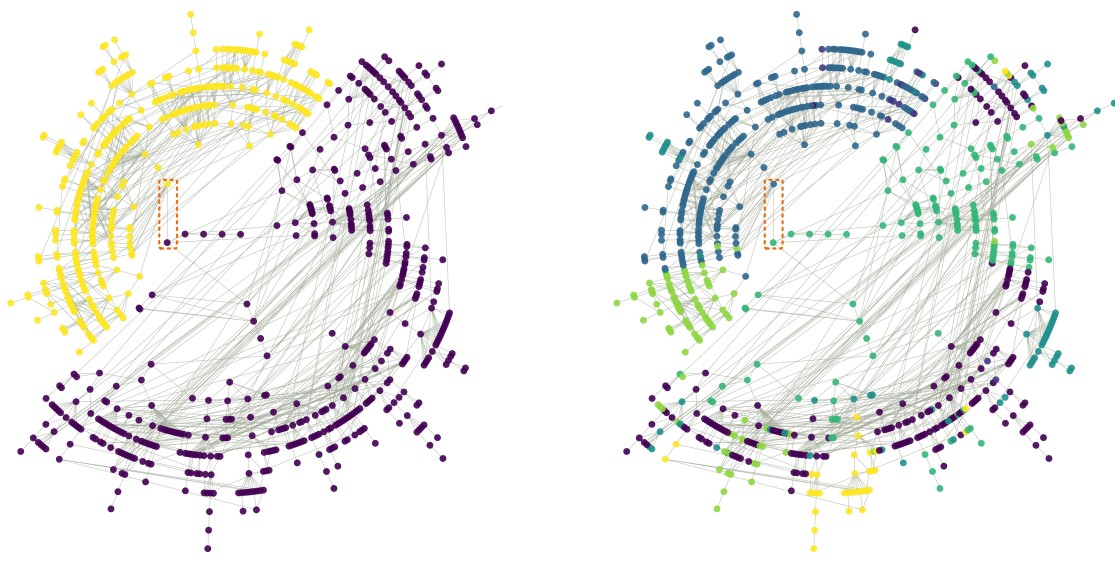

Figure 10: Based on the Cora dataset, we construct a graph with a bottleneck by only having one edge (shown in the orange dash box) linking two different communities. Left: nodes labelled by communities. Right: nodes labelled by true classes.

until only one edge remains, connecting the two communities. This edge can be considered a bottleneck in the graph. The pseudo-code for constructing this graph $\mathcal{G}_o$ is shown in the Algorithm 4.

**Detail of Cora-based Graph.** It is important to consider a dataset's specific characteristics and purpose before making any modifications to it, as it can affect the validity and reliability of the analysis and results obtained from it. We construct a graph with bottlenecks $\mathcal{G}_o$ based on the real dataset **Cora**. The statistics of $\mathcal{G}_o$ are shown in Table 12, and visualization is shown in Figure 10. As Cora is a citation dataset, it consists of a set of nodes representing scientific publications and edges representing citations between them. If some nodes and corresponding edges are deleted from the dataset, it will not affect the authenticity of the dataset as long as it still represents the citation relationships among the remaining publications.

However, not all datasets are suitable for the above constructing method. Take the *PROTEINS* dataset as an example. This dataset represents the interactions between different proteins in a biological system, and the edges in the dataset represent the interactions between them. Suppose some nodes and corresponding edges are deleted from the dataset. In that case, it could potentially alter the overall structure and function of the modelled biological system, thus affecting the authenticity of the dataset.

### A.4 Additional Experiment Results

#### A.4.1 Best Performance of Various GNN Models in Experiment 4.1

We thoroughly examined and tuned the number of layers for each model using the validation set. This has led to a more nuanced understanding of how layer configurations impact model performance. Table 16 shows the highest accuracy achieved by various GNN models across diverse datasets. The table provides insights into whether each model attained its peak performance through a 2-layer, 3-layer, or 4-layer configuration,

Table 16: **Best Performance Accuracy of Various GNN Models Across Datasets.** Most models achieved optimal performance with a 2-layer setting, which is the default and hence not specifically marked. Models achieving their best performance at 3 or 4 layers have this indicated in brackets.

|  | Citeseer | Cora | PubMed | CS | Computers | Photo | ogbn-arxiv |
|---|---|---|---|---|---|---|---|
| GCN | $72.81_{\pm0.41}$ | $80.03_{\pm0.52}$ | $\mathbf{78.15}_{\pm0.52}$ | $90.71_{\pm0.91}$ | $61.47_{\pm3.14}$ | $80.67_{\pm3.59}$ | $71.10^{(3)}_{\pm0.64}$ |
| GATv2 | $72.49_{\pm0.81}$ | $78.36_{\pm1.66}$ | $77.40_{\pm0.49}$ | $89.28_{\pm1.62}$ | $70.98_{\pm3.83}$ | $83.45^{(3)}_{\pm4.52}$ | $\mathbf{71.76}^{(3)}_{\pm0.34}$ |
| SAGE | $71.78_{\pm1.25}$ | $80.19_{\pm0.60}$ | $76.24_{\pm0.51}$ | $90.64_{\pm0.63}$ | $80.75_{\pm0.84}$ | $87.97_{\pm0.48}$ | $71.15^{(4)}_{\pm1.00}$ |
| GIN-$\epsilon$ | $70.12_{\pm1.47}$ | $78.95_{\pm1.02}$ | $77.61_{\pm0.73}$ | $90.80_{\pm0.51}$ | $33.73_{\pm6.25}$ | $65.19_{\pm7.22}$ | $60.95_{\pm1.66}$ |
| AERO | $68.12^{(3)}_{\pm1.78}$ | $\underline{81.79}_{\pm0.93}$ | $74.91^{(3)}_{\pm3.25}$ | $89.06_{\pm1.83}$ | $\mathbf{81.18}_{\pm2.14}$ | $\mathbf{91.59}_{\pm0.97}$ | $71.32^{(3)}_{\pm0.86}$ |
| RGCN | $\underline{73.52}_{\pm1.53}$ | $77.78_{\pm1.06}$ | $75.73^{(3)}_{\pm0.77}$ | $91.13_{\pm0.59}$ | $78.80_{\pm1.33}$ | $85.00_{\pm0.76}$ | $70.45_{\pm0.85}$ |
| MCGCN | $73.44_{\pm1.88}$ | $77.11_{\pm0.30}$ | $76.66^{(3)}_{\pm0.56}$ | $\underline{91.27}_{\pm0.44}$ | $77.78_{\pm1.24}$ | $83.03_{\pm1.62}$ | $69.67_{\pm1.21}$ |
| PGCN | $73.24_{\pm2.05}$ | $77.78_{\pm0.99}$ | $76.35^{(3)}_{\pm0.68}$ | $88.30_{\pm0.58}$ | $78.39_{\pm1.49}$ | $86.48_{\pm1.76}$ | $70.53_{\pm0.76}$ |
| D2GCN | $73.20_{\pm0.70}$ | $80.41_{\pm0.54}$ | $77.84_{\pm0.71}$ | $90.46_{\pm0.58}$ | $80.02_{\pm1.74}$ | $87.03_{\pm0.80}$ | $70.61_{\pm0.26}$ |
| **LDGCN** | $\mathbf{74.33}_{\pm0.79}$ | $\mathbf{81.93}^{(3)}_{\pm1.40}$ | $\underline{78.02}_{\pm0.34}$ | $\mathbf{91.53}_{\pm0.41}$ | $\underline{80.81}_{\pm0.26}$ | $88.84^{(3)}_{\pm1.48}$ | $\underline{71.66}^{(4)}_{\pm0.30}$ |

denoted within brackets where applicable. In the table, the **best-performing** method for each dataset is highlighted in bold, emphasizing the top achievement, while the second-best performance is underscored for clarity.

The results revealed that while a 2-layer setup is generally effective, certain models and datasets benefit from more layers. For instance, our LDGCN exhibited its highest accuracies on Cora and Photo datasets with 3 layers. An interesting pattern emerged from our analysis: methods that incorporated negative samples frequently achieved the second-best results. This observation suggests that adding negative samples to graph convolutional neural networks can significantly enhance the model's ability to learn different types of relationships, thereby improving overall performance. However, it's crucial to note that not all methods involving negative samples yielded consistent performance across different datasets. This variability highlights that the selection of negative samples is a non-trivial aspect of model design and that carefully chosen negative samples can substantially aid GCNs in better learning from graph data.

### A.4.2 Big Dataset

We conducted additional experiments using the Open Graph Benchmark's login-mag dataset, which indeed presents a more challenging environment with its larger graph size of approximately 1.94 million nodes and over 21 million edges. The statistics of obgn-mag are shown in Table 17.

Table 17: STATISTICS of obgn-mag

| Dataset | Nodes | Edges | Classes | Average of Degree | Split / Ration | Epoch |
|---|---|---|---|---|---|---|
| ogbn-mag | 1,939,743 | 21,111,007 | 349 | 21.7 | 85/9/6 | 400 |

The eigendecomposition techniques employed in DPP sampling do introduce considerable computational complexity, which can lead to scaling challenges on very large graphs. To address this and maintain a balance between computational efficiency and the benefits of our approach, we strategically sampled a subset of 0.1% of the nodes from the ogbn-mag dataset for layer-diverse negative sampling. The results are shown in Table 18.

Despite the reduced sampling size, our LDGCN achieved an accuracy of $33.51\%_{\pm0.32}$ on the ogbn-mag dataset. This performance is higher compared to the baseline models. The results demonstrate that even with only a fraction of the nodes subjected to layer-diverse negative sampling, there is still enhancement

in the performance of the original GCN framework. This evidences the efficacy of our proposed sampling method, suggesting that it could be a valuable strategy for managing the trade-off between computational demand and performance in large-scale graph neural networks.

Table 18: Acc of 4-layer models on obgn-mag dataset

|  | GCN | GATv2 | GraphSAGE | LDGCN |
|---|---|---|---|---|
| obgn-mag | $31.99_{\pm 0.53}$ | $32.21_{\pm 0.46}$ | $30.11_{\pm 0.29}$ | $33.51_{\pm 0.32}$ |

### A.4.3 Graphs Classification

We conducted additional experiments focusing on graph-level classification tasks, this time extending our approach to the Graph Isomorphism Network (GIN) model. We selected two well-known graph datasets, *Proteins* and *MUTAG*, for our experiments. We employed a 10-fold cross-validation scheme and utilized SUM readout pooling to aggregate node features at the graph level. Our experiments compared the performance of standard graph neural network models like GCN, GATv2, GraphSAGE, and GIN with our modified version, the Layer-Diverse GIN (LD-GIN). The results of these experiments are shown in the Table 19.

Table 19: Acc of 2-layer models on graph dataset

|  | GCN | GATv2 | GraphSAGE | GIN | LD-GIN |
|---|---|---|---|---|---|
| PROTEINS | $72.97_{\pm 2.55}$ | $64.11_{\pm 7.19}$ | $72.43_{\pm 1.57}$ | $73.06_{\pm 2.14}$ | $\mathbf{74.07}_{\pm 4.78}$ |
| MUTAG | $76.54_{\pm 8.19}$ | $77.78_{\pm 6.92}$ | $79.16_{\pm 4.60}$ | $87.83_{\pm 4.89}$ | $\mathbf{88.89}_{\pm 6.05}$ |

The results from these experiments were encouraging. Our LD-GIN model improved over the baseline models on both the *PROTEINS* and *MUTAG* datasets. This enhanced performance on graph-level classification tasks demonstrates the versatility of our layer-diverse negative sampling method. It not only maintains its effectiveness across different graph structures but also adapts to varying task requirements, whether they involve node-level or graph-level classification. This extension of our experiments to graph-level tasks aligns to broaden the applicability and relevance of our method.

