# OpenReview forum: "Layer-diverse Negative Sampling for Graph Neural Networks"
_TMLR — Accepted by TMLR_

### Review · Reviewer_qXu9 · 2023-10-23

**Summary Of Contributions:**

This paper proposes a layer-diverse negative sampling method for graph neural networks (GNNs) to mitigate issues like over-smoothing and over-squashing. Traditional GNNs only gather information from first-order neighbors (positive samples) during message passing. This can lead to limited expressivity and over-smoothing. Negative sampling incorporates non-neighboring nodes (with distinct representations) into message passing. This enhances representation learning. Existing negative sampling methods treat samples independently per layer. This results in redundancy across layers. The proposed method transforms the candidate set into a space using a sampling matrix in determinantal point processes (DPP). It then squeezes this space along dimensions corresponding to previous layer's samples. This reduces overlap. Experiments on 7 datasets show consistently superior performance over baselines. Layer-diverse sampling reduces node and cluster overlap versus prior DPP sampling. Negative sampling can improve GNN expressivity by temporarily changing graph topology. It can also alleviate over-squashing by introducing new negative edges without over-smoothing.

**Audience:**

Yes

**Claims And Evidence:**

Yes

**Requested Changes:**

* It would be more convincing to add some case studies.

* It would be more convincing to add results on some larger datasets.

**Strengths And Weaknesses:**

### Strengths:

* Effective novel method for negative sampling in GNNs

The paper presents a new way to do negative sampling in graph neural networks using determinantal point processes and space squeezing. This is an innovative technique not explored before. Experimental results demonstrate that this novel approach consistently outperforms existing negative sampling methods and baseline GNNs without sampling across diverse datasets. So the strength is proposing and validating a new effective technique to enhance representation learning in GNNs.

* Reduces redundancy across layers

Prior negative sampling methods treat each layer independently, leading to high redundancy in samples across layers. The proposed space squeezing method significantly reduces overlap in negative samples between consecutive layers. This is empirically shown through node and cluster overlap metrics. Lower overlap implies more useful information from sampling, avoiding repeated nodes across layers.

* Applicable to different GNN architectures

The layer-diverse negative sampling technique is tested on various GNN models like GCN, GAT, GraphSAGE and shows consistent improvements in their performance. This demonstrates the wide applicability and flexibility of the proposed approach across diverse architectures beyond a single model.

### Weaknesses

* Computational complexity

The eigendecomposition and space squeezing in DPP sampling add significant computational overhead. This causes scaling issues for very large graphs with millions of nodes and edges. Approximate solutions would be needed. The complexity also increases with more layers in the GNN, limiting depth of models.

* Universal benefits not guaranteed

While the technique helps most models on most datasets, the GIN architecture failed to benefit on the Computers dataset. This highlights dataset-specific and model-specific factors may affect gains seen from the approach. More analysis needed on where the technique consistently falls short.

* Lack of case studies or qualitative results

The paper lacks any case studies or qualitative results to provide better intuition about the model's effectiveness. For example, visualizing the differences in negative samples selected by the proposed method versus prior techniques could have helped build better intuition. Providing some examples demonstrating improved representation learning through case studies would have been beneficial.

* Small datasets used in experiments

All the datasets used in the experiments are relatively small. The model should have been evaluated on some larger graphs with millions of nodes to better demonstrate scalability. Using larger datasets would have strengthened the empirical results and conclusions drawn.

---

> ### Author Response · Authors · 2024-01-18
> **Thank you so much for your constructive comments, please see the detailed point-by-point response below.**
>
> > It would be more convincing to add some case studies.
>
>  Thank you so much for your constructive comments. We agree that such insights could significantly enhance the intuitive understanding of our model's effectiveness. In response to this comment, we have conducted a case study using the Cora dataset to qualitatively compare our proposed LDGCN and D2GCN. **The results are shown in Fig.9. Please check Sec.4.6 in the revised version.**
>
> > It would be more convincing to add results on some larger datasets.
>
> We are grateful for your recommendation to test our model on larger datasets to demonstrate its scalability and effectiveness better. In line with your suggestion, we have included additional experiments using the Open Graph Benchmark's login-mag dataset, which indeed presents a more challenging environment with its larger graph size of approximately 1.94 million nodes and over 21 million edges.
>
> **Acc of 4-layer models on obgn-mag dataset**
>
> |           | GCN            | GATv2          | GraphSAGE      | LDGCN          |
> |-----------|----------------|----------------|----------------|----------------|
> | obgn-mag  | 31.99 ±0.53    | 32.21 ±0.46    | 30.11 ±0.29    | 33.51 ±0.32    |
>
> We acknowledge that the eigendecomposition techniques employed in Determinantal Point Process (DPP) sampling do introduce considerable computational complexity, which can lead to scaling challenges on very large graphs. To address this and maintain a balance between computational efficiency and the benefits of our approach, we strategically sampled a subset of 0.1\% of the nodes from the ogbn-mag dataset for layer-diverse negative sampling. The results are shown in the above table.
>
> Despite the reduced sampling size, our Layer-Diverse Graph Convolutional Network (LDGCN) achieved an accuracy of 33.51 ±0.32   on the ogbn-mag dataset. This performance is higher compared to the baseline models. The results demonstrate that even with only a fraction of the nodes subjected to layer-diverse negative sampling, there is still enhancement in the performance of the original GCN framework. This evidences the efficacy of our proposed sampling method, suggesting that it could be a valuable strategy for managing the trade-off between computational demand and performance in large-scale graph neural networks.
>
> Furthermore, we are actively working on finding more effective approximate solutions that will allow us to apply layer-diverse negative sampling to even larger datasets while managing computational resources efficiently.

---

### Review · Reviewer_qV4F · 2023-11-07

**Summary Of Contributions:**

This paper introduces a novel approach to mitigating over-smoothing and over-squashing issues in traditional GNNs through the incorporation of layer-diverse negative sampling using a determinantal point process and a space-squeezing method in multi-layer GNNs. The experimental results demonstrate better performance on node classification tasks.

**Audience:**

Yes

**Broader Impact Concerns:**

No ethical concern.

**Claims And Evidence:**

No

**Requested Changes:**

1. Provide more rigorous theoretical analyses, especially for expressiveness and over-squashing.
2. Enhance the experimental study with more convincing results.
3. Conduct extended experiments for applicability exploration.

**Strengths And Weaknesses:**

Strengths:
1. The utilization of negative sampling to address over-smoothing and over-squashing in GNNs is conceptually sound and intuitive.
2. The paper offers a comprehensive presentation of the problem, methodology, analysis, and experimental results.

Weaknesses:
1. Lack of Thorough Theoretical Analysis:
   - It is essential to provide a more rigorous theoretical foundation to support the claim of improved expressiveness. It is insufficient to claim better expressiveness by providing several good cases. It is easy to find some bad cases, which are distinguishable by GNN but become indistinguishable by using negative sampling.
   - The analysis of over-squashing is not convincing. Existing literature indicates that the over-squashing can be remitted by adding connections between two dense clusters. It is not clear if adding negative connections can also help. Moreover, there is no evidence to show that negative sampling is more focused within dense clusters or across clusters. More theoretical evaluation is needed.

2. Doubtful Experimental Results:
   -  In Figure 7, the model performance with the number of layers set to 4 never achieves the best when compared against other number of layers. However, Table 1 only reports the results at 4 layers, which is unrepresentative. Better to tune the number of layers by using the validation set and report the best performance.
   - The baseline results for the Computers and Photo datasets appear significantly lower than the results reported on the leaderboard. The performance of GCN on Computers and Photo are always higher than 80% and 90%, but only 47% and 68% in this paper. This raises concerns about the reproducibility and reliability of the reported results. A more rigorous comparison with established baselines is necessary.

3. Limited Exploration of Applicability:
   - It would be beneficial to conduct extensive comparisons by incorporating the proposed method with state-of-the-art methods to assess the potential for further improvement.
   - Additionally, investigating whether the proposed method is applicable to graph-level tasks is an avenue worth exploring, as it can enhance the paper's applicability and relevance to a broader range of problems.

---

> ### Author Response · Authors · 2024-01-18
> **Thank you so much for your constructive comments, please see the detailed point-by-point response below.**
>
> > Lack of Thorough Theoretical Analysis
>
> Thank you for pointing out the necessity of a more rigorous theoretical analysis in our manuscript, particularly concerning the expressiveness and over-squashing aspects of our proposed method. We understand the importance of underpinning our empirical findings with strong theoretical foundations, and we appreciate your feedback in this regard.
>
> We acknowledge that developing a comprehensive theoretical analysis, especially in the areas highlighted, is a substantial and intricate endeavour. Given the complexity of these topics and the depth of analysis required, providing a thorough theoretical framework within the short timeframe of two weeks presents significant challenges.
>
> We recognize the value and importance of this aspect of our research. We are committed to continuing our work in this direction and plan to develop and include more robust theoretical analyses in future iterations of our work. This will be a focal point of our ongoing research efforts, as we aim to provide a deeper understanding of the theoretical underpinnings that support the empirical results presented.
>
> > Doubtful Experimental Results. Enhance the experimental study with more convincing results.
>
> Thanks for your comments. We appreciate your constructive feedback and have taken steps to address the concerns raised.
>
>  **1.Tuning the Number of Layers**
>
> In response to your suggestion, we thoroughly examined and tuned the number of layers for each model using the validation set. This has led to a more nuanced understanding of how layer configurations impact model performance. The Tab.1 presents these results.
>
> |             | Citeseer         | Cora             | PubMed           | CS               | Computers        | Photo            | ogbn-arxiv       |
> |-------------|------------------|------------------|------------------|------------------|------------------|------------------|------------------|
> | GCN         | 72.81 ±0.41      | 80.03 ±0.52      | **78.15 ±0.52**  | 90.71 ±0.91      | 61.47 ±3.14      | 80.67 ±3.59      | 71.10 ±0.64(3)    |
> | GATv2       | 72.49 ±0.81      | 78.36 ±1.66      | 77.40 ±0.49      | 89.28 ±1.62      | 70.98 ±3.83      | 83.45 ±4.52(3)    | **71.76 ±0.34**(3)|
> | SAGE        | 71.78 ±1.25      | 80.19 ±0.60      | 76.24 ±0.51      | 90.64 ±0.63      | 80.75 ±0.84    | 87.97 ±0.48    | 71.15 ±1.00(4)    |
> | GIN-ε       | 70.12 ±1.47      | 78.95 ±1.02      | 77.61 ±0.73      | 90.80 ±0.51      | 33.73 ±6.25      | 65.19 ±7.22      | 60.95 ±1.66      |
> | RGCN        | 73.52 ±1.53    | 77.78 ±1.06      | 75.73 ±0.77(3)    | 91.13 ±0.59      | 78.80 ±1.33      | 85.00 ±0.76      | 70.45 ±0.85      |
> | MCGCN       | 73.44 ±1.88      | 77.11 ±0.30      | 76.66 ±0.56(3)    | 91.27 ±0.44    | 77.78 ±1.24      | 83.03 ±1.62      | 69.67 ±1.21      |
> | PGCN        | 73.24 ±2.05      | 77.78 ±0.99      | 76.35 ±0.68(3)    | 88.30 ±0.58      | 78.39 ±1.49      | 86.48 ±1.76      | 70.53 ±0.76      |
> | D2GCN       | 73.20 ±0.70      | _80.41 ±0.54_    | 77.84 ±0.71      | 90.46 ±0.58      | 80.02 ±1.74      | 87.03 ±0.80      | 70.61 ±0.26      |
> | **LDGCN**   | **74.33 ±0.79**  | **81.93 ±1.40**(3)|  78.02 ±0.34   | **91.53 ±0.41**  | **80.81 ±0.26**  | **88.84 ±1.48**(3)| 71.66 ±0.30(4)  |
>
>
> Importantly, the table specifies whether the best performance for each model was achieved with a 2-layer, 3-layer, or 4-layer configuration, as indicated in brackets where applicable. This approach revealed that while a 2-layer setup is generally effective, certain models and datasets benefit from a higher number of layers. For instance, our LDGCN exhibited its highest accuracies on Cora and Photo datasets with 3 layers.
>
>  An interesting pattern emerged from our analysis: methods incorporating negative samples frequently achieved the second-best results. This observation suggests that adding negative samples to graph convolutional neural networks can significantly enhance the model's ability to learn different types of relationships, thereby improving overall performance. However, it's crucial to note that not all methods involving negative samples yielded consistent performance across different datasets. This variability highlights that the selection of negative samples is a non-trivial aspect of model design and that carefully chosen negative samples can substantially aid GCNs in better learning from graph data.
>
>  We are grateful for the opportunity to improve our work based on your feedback. We hope these additional results will satisfactorily address your comments regarding the representation of model performance.

---

> ### Author Response · Authors · 2024-01-18
> **Response continued**
>
> > Baseline Comparison on Amazon Datasets
>
> Regarding the baseline results for the Computers and Photo datasets, we have conducted a detailed analysis to understand the differences between our reported results and those on the leaderboard. One significant deviation between our implementation and those methods reported on the leaderboard is the use of layer normalization. Our approach in constructing the models was to adhere closely to the settings described in the original GCN [1], which did not employ any form of normalization between layers. This difference in methodology is crucial to note as it impacts the comparison of results.
>
> Despite not using normalization in our models, our Layer-Diverse Graph Convolutional Network (LDGCN) demonstrated commendable performance in Computers and Photo datasets. Moreover, as evident from Table 1 in our submission and the best performance Table shown above, the integration of negative samples into the GCN framework, irrespective of the sampling method, led to performance improvements on these datasets. This observation strongly implies that incorporating negative samples into the graph convolution operation benefits graph learning. These findings underscore the value of negative sampling in enhancing the capabilities of GCN models, even in the absence of layer normalization.
>
> [1]Thomas N. Kipf, Max Welling: Semi-Supervised Classification with Graph Convolutional Networks. ICLR (Poster) 2017
>
> > Limited Exploration of Applicability
>
> Thank you so much for your constructive comments. We recognize the importance of demonstrating the versatility and broad applicability of our proposed layer-diverse negative sampling method. In response, we have conducted additional experiments focusing on graph-level classification tasks, this time extending our approach to the Graph Isomorphism Network (GIN) model.
>
> We selected two well-known graph datasets, *Proteins* and *MUTAG*, for our experiments. We employed a 10-fold cross-validation scheme and utilized SUM readout pooling to aggregate node features at the graph level. Our experiments compared the performance of standard graph neural network models like GCN, GATv2, GraphSAGE, and GIN with our modified version, the Layer-Diverse GIN (LD-GIN). The results of these experiments are shown in the following table:
>
> **Table: Acc of 2-layer models on graph dataset**
>
> |           | GCN             | GATv2           | GraphSAGE       | GIN             | LD-GIN          |
> |-----------|-----------------|-----------------|-----------------|-----------------|-----------------|
> | PROTEINS  | 72.97 ±2.55     | 64.11 ±7.19     | 72.43 ±1.57     | 73.06 ±2.14     | 74.07 ±4.78     |
> | MUTAG     | 76.54 ±8.19     | 77.78 ±6.92     | 79.16 ±4.60     | 87.83 ±4.89     | **88.89 ±6.05** |
>
> The results from these experiments were encouraging. LD-GIN model improved over the baseline models on both the *PROTEINS* and *MUTAG* datasets. This enhanced performance on graph-level classification tasks demonstrates the versatility of our layer-diverse negative sampling method. It not only maintains its effectiveness in different graph structures but also adapts to varying task requirements, be it node-level or graph-level classification.
>
> This extension of our experiments to graph-level tasks aligns to broaden the applicability and relevance of our method. We believe that these additional findings substantially enrich the manuscript.

---

### Review · Reviewer_3kP1 · 2024-01-04

**Summary Of Contributions:**

This paper proposes a new negative sampling methods to improve the message passing of graph neural networks. Negative samples are found to be useful to prevent oversmoothing and overquashing in previous works; however the previous DPP-based sampling methods contain too much redundancy in different layers. This paper improves the approach in (Duan et al. 2022) by advancing a new layer-diverse DPP sampling schema. Experiments show that it can reduce the overlap rates of negative samples and improve the accuracy of GNNs.

**Audience:**

Yes

**Claims And Evidence:**

Yes

**Requested Changes:**

1. Revising the paper, correcting the typos, and trying to make Sec2 easier to follow. Example typos: Page5, "the hope is reduce the information of any" --> "the goal is to reduce the information of any"
2. Adding experiments on some heterophilous graphs such as Cornell, Texas, and Wisconsin.

**Strengths And Weaknesses:**

Strengths:
1. It is an interesting and reasonable extension to improve D2GCN(Duan et al. 2022). Table 2 and 3 are good proof of the motivation.
2. It is good to discuss how it impacts expressivity and oversquashing and experimentally demonstrate it.

Weakness:
1. When I first read it, my impression is it is not self-contained and thus hard to follow. However after checking the paper of D2GCN, I found that the formulation and background are very similar. So, maybe it is just the writing issue. For readers who are not familiar with DPP, it may take some time for they to understand the whole flow of how you define L and how you use it to do sampling. I believe Section 2 can be improved to make it easier understandable (e.g. by adding some simple description of the whole pipeline).
2. The major issue is the practicality. As the authors may already notice, L is different for every node; the procedures of candidate set generation/eigendecomposition/negative sampling are time consuming compared to message passing. Although only sampling 1% of the nodes can still show improvement, the scalability is still a serious issue when the graph is larger.
3. The paper focused more on the discussion of oversmoothing and oversquashing, but I think it is more related to heterophily. In the case of heterophilious graphs, intuitively there are more negative samples and negative sampling is more useful. Unfortunately the paper did not have experiments on heterophilous graphs. It is better to add experiments on those datasets.
4. Following 3 and 2, on heterophilous graphs, there are many advanced algorithms which are not only effective but also efficient, such as GCNII. I checked the leaderboard of ogb-arxiv, GCNII is also better than the reported results in this paper. Considering there are easily alternatives to be more efficient while having similar or even better results, the usefulness of the proposed negative sampling method is doubtful.

---

> ### Author Response · Authors · 2024-01-18
> **Thank you so much for your constructive comments, please see the detailed point-by-point response below.**
>
> > Adding experiments on some heterophilous graphs such as Cornell, Texas, and Wisconsin.
>
> Thank you so much for your constructive comments. Regarding exploring our model's effectiveness in the context of heterophilous graphs. You pointed out that heterophily in graphs could naturally provide more negative samples, potentially amplifying the advantages of negative sampling. Following your recommendation, we acknowledge this aspect and have expanded our experiments to include heterophilous datasets such as Cornell, Texas, and Wisconsin (collectively referred to as the WebKB dataset).
>
> To ensure a comprehensive analysis, we tested the layer-diverse negative sampling method across multiple graph neural network architectures, both in 2-layer and 4-layer configurations. The architectures tested include GCN, GATv2, GraphSAGE, and GIN.
>
> The results of the 2-layer are shown in Tab.1. On the Cornell dataset, our LD-GCN model outperformed the standard GCN by approximately 6.84\%. In Texas, the LD-GATv2 model showed an improvement of 11.32\% over the standard GATv2. For Wisconsin, LD-SAGE exceeded the performance of standard GraphSAGE by 5.82\%.
>
> Furthermore, the 4-layer model results (Tab.2) are consistent with the improved performance observed in the 2-layer models, suggesting that our layer-diverse negative sampling method contributes positively across different model depths.
>
> Table 1: Acc of 2-layer models on WebKB dataset
>
> |            | Cornell        | Texas          | Wisconsin      |
> |------------|----------------|----------------|----------------|
> | GCN        | 48.52 ±5.09    | 56.21 ±5.65    | 49.80 ±5.70    |
> | LD-GCN     | 55.36 ±6.04    | 61.62 ±5.90    | 61.56 ±5.63    |
> | GATv2      | 51.35 ±7.15    | 50.54 ±4.21    | 50.54 ±4.21    |
> | LD-GATv2   | 66.48 ±4.71    | 61.86 ±7.36    | 64.31 ±6.72    |
> | GraphSAGE  | 61.01 ±4.17    | 70.27 ±5.04    | 70.65 ±2.86    |
> | LD-SAGE    | 67.11 ±7.54    | 76.46 ±4.52    | 76.47 ±6.32    |
> | GIN        | 43.78 ±4.49    | 56.48 ±5.19    | 47.05 ±5.33    |
> | LD-GIN     | 56.78 ±3.05    | 61.56 ±5.37    | 52.94 ±4.16    |
>
>
> Table 2: Acc of 4-layer models on WebKB dataset
>
> |            | Cornell        | Texas          | Wisconsin      |
> |------------|----------------|----------------|----------------|
> | GCN        | 43.78 ±6.70    | 54.23 ±6.52    | 53.13 ±6.92    |
> | LD-GCN     | 48.65 ±5.37    | 58.64 ±5.42    | 58.47 ±6.02    |
> | GATv2      | 49.54 ±8.50    | 57.97 ±6.33    | 51.52 ±5.37    |
> | LD-GATv2   | 52.54 ±3.86    | 60.06 ±3.78    | 57.14 ±3.54    |
> | GraphSAGE  | 52.97 ±6.41    | 64.86 ±5.40    | 59.37 ±5.28    |
> | LD-SAGE    | 58.59 ±6.10    | 70.64 ±7.61    | 62.94 ±7.21    |
> | GIN        | 48.38 ±7.29    | 58.39 ±4.56    | 47.12 ±4.43    |
> | LD-GIN     | 54.05 ±4.69    | 60.48 ±4.03    | 54.37 ±3.15    |
>
> Heterophilous graphs are characterized by their tendency to connect nodes with dissimilar features or labels. This starkly contrasts the homophilous nature typically assumed in many graph neural network (GNN) designs. This heterophily implies a diverse neighbourhood for each node, which can challenge learning algorithms that rely on the assumption that 'neighbouring nodes have similar labels or features'.
>
> Our layer-diverse negative sampling method is well-suited for such graphs for several reasons:
>
> - **DPP-based Sampling Within Layers**: Our method uses DPP-based sampling to ensure diversity within each layer of the graph. This approach is crucial for heterophilous graphs, where it's important to capture a wide range of node characteristics within the same layer.
>
> - **Layer-Diverse Enhancement:** We enhance diversity between layers and reduce overlap, allowing for richer information to be captured across the graph. This method is particularly effective in heterophilous graphs, where nodes with similar properties may not be close in the graph's topology.
>
> - **Improved Node Representation Learning:** Our approach effectively learns node representations by distinguishing between similar and dissimilar neighbours. This is key in heterophilous graphs, where traditional GNNs might struggle due to the uniformity in their aggregation and update processes.
>
> - **Structural Insight:** Our method offers more structural insight into the graph by allowing the GNN to learn from a wider range of node connections, thus avoiding the pitfall of homogeneity in the learning process.
>
> We believe that these results and our analysis of the structural properties of heterophilous graphs demonstrate the applicability and advantages of our layer-diverse negative sampling method in a broader range of graph types. This strengthens the case for our approach as a versatile tool in the GNN toolkit, capable of addressing the challenges presented by both homophilous and heterophilous graphs.

---

> ### Author Response · Authors · 2024-01-18
> **Response continued**
>
> > The scalability is still a serious issue when the graph is larger.
>
>  Thank you so much for your constructive comments. To further test the scalability of our model, We have included additional experiments using the Open Graph Benchmark's login-mag dataset, which indeed presents a more challenging environment with its larger graph size of approximately 1.94 million nodes and over 21 million edges. This performance of LDGCN is higher compared to the baseline models. Please refer to the comments to **Reviewer qXu9**  for more details.
>
> > The usefulness of the proposed negative sampling method is doubtful.
>
> Our proposed method centred around the concept of layer-diverse negative sampling, is specifically designed to address a different aspect of graph neural network (GNN) performance. The primary objective of our approach is to enrich the learning process by sampling meaningful and diverse negative samples for each node. This method aims to provide more varied and comprehensive information during the graph convolution operation, a feature that is not the central focus of models like GCNII.
>
> The unique value of our method lies in its ability to enhance the expressiveness of GNNs by capturing a broader spectrum of node relationships, which is critical in complex graph structures. While efficiency and overall effectiveness are important metrics, our method contributes to the diversity and quality of the learned representations. This aspect is particularly crucial in scenarios where the diversity of sampled information plays a key role in the model's performance, like on heterophilous graphs.

---

### Review · Reviewer_kkpA · 2024-01-05

**Summary Of Contributions:**

This paper, targeting the over-smoothing and over-squashing problem, proposes a solution to add negative samples (nodes) and alter the original graph topology. Specifically, the proposed method aims to sample the negative samples (nodes) in an effecient way, and then add an additional term as the minus of the summation of the negative nodes (Equation 4).

**Audience:**

Yes

**Claims And Evidence:**

No

**Requested Changes:**

As detailed in the weakness and strengths part, the recommended revisions are:

1. Explanations on the motivation

2. Justification of the formulation in equation 4

3. Discussion and comparison with new works that focus on the same problem (over-smoothing and over-squashing)

**Strengths And Weaknesses:**

The proposed method can enhance the performance of several existing GNNs according to experiments, however, there are some major concerns listed below.

The formulation of equation 4 is a core part of the work, however, this formulation lacks explanation. Why is the negative nodes simply minused from the message passing formulation? It seems to imply that the negative nodes contain irrelavant information and should be deleted. However, the original term consisting of the positive node aggregation (i.e. the first term of equation 4) does not contain the information from the negative samples, therefore this design is not convincing and require serious justification.

While the aim to resolve the over-smoothing and over-squashing problems is meaningful, the explanations on how the proposed method is related to this target problem is not satisfying, with some vague claims. For example, in the second paragraph of Introduction, since the aim of GNN is to refine the representation of a given node with its neighbors with similar information, why would it be important to incorporate distant 'negative nodes' with very different information? Node representations are local, unlike the graph representation of the entire graph, and it is not straightforwardly convincing to say that each node should contain distant information, which may blur the node's own information. Moreover, it is also unclear what is the relation between the negative sampling and the over-smoothing/squashing problem, which is the motivation of this work and should be clarified at the very beginning.

The proposed method is applied to different GNN models in experiments. However, the adopted GNNs are mostly old ones without consideration on the over-smoothing or over-squashing problem. Therefore, it is necessary to discuss what the advantages of the proposed method are, being compared to other works also studying over-smoothing and over-squashng problem, both conceptually and experimentally.

---

> ### Author Response · Authors · 2024-01-18
> **Thank you so much for your constructive comments, please see the detailed point-by-point response below.**
>
> > Justification of the formulation in equation 4.
>
> Thanks for the comments! Firstly, you are right that "the negative nodes contain irrelevant information and should be deleted." In the original message passing framework (using only the first term of Eq. 4), a node will aggregate the information from its neighbours (positive nodes). Note that these neighbours contain the information from other nodes, including the negative samples due to the message passing. After rounds of message passing, this node (or its feature representation) will also contain the information from other nodes, including negative samples. Indeed, it is the underlying reason why the over-smoothing happens where all nodes tend to have similar representations. Our equation 4 is designed to resolve this problem by explicitly "removing the information of negative samples during the aggregation and message passing", as understood by the reviewer. As suggested by the reviewer, we will add these discussions after the equation in the final version of the paper.
>
> > Discussion and comparison with new works that focus on the same problem (over-smoothing and over-squashing).
>
> Thanks for the comments! In the original message-passing framework of GNNs, a node will aggregate the information from its neighbours (positive nodes). Note that these neighbours contain the information from other nodes, including the negative samples due to the message passing. After rounds of message passing, this node (or its feature representation) will also contain the information from other nodes, including negative samples. Indeed, it is just the underlying reason why the over-smoothing happens, where all nodes tend to have similar representations after rounds of message passing. Hence, please note that, even without negative samples, the node (or its feature representation) will contain the information from 'distant' nodes after the message passing. The basic idea of introducing negative samples is to restrict the nodes from becoming too similar to each other. In our newly designed new message passing (equation 4), the aggregation of positive nodes (neighbours) will pull a node to be close to others (both neighbours and others), while the aggregation of negative nodes will push this node to be far from the distant (negative) nodes. In such cases, the final embeddings of all nodes will be not too similar to each other, and the over-smoothing is reduced. We have more discussions on that in Section 3.2, and we will add these discussions to the Introduction to let the readers better understand the role of negative samples at the very beginning, as suggested by the reviewer.
>
> We also want to highlight that even though incorporating information from distant nodes may blur the node's own information, it is still worth it with careful design. One example is the rewiring methods [1], which is the pioneer study of the over-squashing problem and also adds new links to the graph in their model. The studies [3] on using negative samples for GNNs (including ours) also show that incorporating information from distant nodes is not bad for the node's own information but beneficial to the final performance.
>
> As suggested, we have compared two specific methods for over-smoothing and over-squashing problems: MAD [3] for over-smoothing and FA [1] for over-squashing. The results of over-smoothing are given below. The results of over-squashing are in the Tab.10 of the submitted paper. We can see that our methods could achieve better performance on two problems.
>
> |          | Citeseer           | Cora               | PubMed             | Coauthor-CS        | Computers          | Photo              | ogbn-arxiv         |
> |----------|--------------------|--------------------|--------------------|--------------------|--------------------|--------------------|--------------------|
> | GCN      | 55.78 ±5.69        | 63.39 ±7.92        | 72.24 ±4.34        | 54.00 ±3.69        | 47.21 ±6.22        | 68.04 ±6.37        | 70.57 ±1.02        |
> | MAD      | 57.14 ±4.27        | 65.73 ±6.37        | 73.89 ±3.91        | 71.80 ±0.72        | 47.50 ±2.89        | 68.99 ±7.75        | 70.89 ±0.87        |
> | **LDGCN**| **68.27 ±1.29**    | **76.80 ±1.26**    | **77.07 ±1.23**    | **86.23 ±0.55**    | **77.92 ±2.34**    | **86.50 ±1.48**    | **71.66 ±0.30**    |
>
>
> [1]. Uri Alon and Eran Yahav. On the bottleneck of graph neural networks and its practical implications.(ICLR 2021), Virtual Event, Austria, 2021
>
> [2]. Zhen Yang, Ming Ding, Chang Zhou, Hongxia Yang, Jingren Zhou, and Jie Tang. Understanding negative sampling in graph representation learning.  (KDD 2020), Virtual Event, CA, USA, August 23-27, pp. 1666–1676, 2020
>
> [3]. Deli Chen, Yankai Lin, Wei Li, Peng Li, Jie Zhou, and Xu Sun. Measuring and relieving the over-smoothing problem for graph neural networks from the topological view. (AAAI 2020), New York, NY, USA, February 7-12, pp. 3438–3445, 2020

---

### Decision · Action_Editor_sQtX · 2024-02-12

**Recommendation:** Accept with minor revision

**Comment:**

Reviewers agreed that the paper contributes to an important topic of research (GNN's issues on over-smoothing and over-squashing). They are satisfied overall with the empirical evidence for justifying the proposed approach.

However, they also pointed out a few issues:
1. Theoretical justification of the proposed approach is lacking;
2. The baseline methods in comparison are not from state-of-the-art (2021 or earlier);
3. The paper's notations and writing are not self-contained; also the introduction to DPP are not easy to digest.

While I believe the authors were not tackling the theoretical challenges in this paper, I believe the rest of the two points need to be addressed in the camera ready version. Hence I recommend the following revisions:
1. Include at least one baseline from the latest paper, or justify explicit at the beginning of the experiment section why some of the SOTA methods won't address the over-smoothing/over-squashing issue;
2. Consider making the paper self-contained by e.g., adding appendix materials to explain the notation/setting better, as well as a longer intro to DPP.

**Audience:**

GNN research community, as well as people interested in determinantal point processes.

**Claims And Evidence:**

This paper studies the over-smoothing and over-squashing issues in GNNs which are important topics in GNN research. The proposed solution mainly uses the idea of determinantal point process to perform negative sampling for the node data, in order to improve the diversity of messages in the message passing process. The proposed approach is mainly empirically justified with a large number of experiments.

---

> ### Author Response · Authors · 2024-03-11
> **Thank you for your insightful feedback on our manuscript. Below, we outline the revisions we have made in response to your suggestions:**
>
> 1.**Baseline Inclusion and Experimental Justification**: We have included a SOTA method named AERO (ICML 2023) [1] in our experiments. The corresponding results have been updated in Table 1, Table 16, and Figure 7 to provide a comprehensive comparison and demonstrate the effectiveness of our proposed approach.
>
> 2.**Revision for Preliminaries and Appendix**: To enhance the self-containment of our paper, we have revised the Preliminaries section to make it more readable. Additionally, we have added more introductions about DPP in the Appendix to provide readers with a deeper understanding of the technical aspects of our methodology.
>
> [1] Soo Yong Lee, Fanchen Bu, Jaemin Yoo, Kijung Shin: Towards Deep Attention in Graph Neural Networks: Problems and Remedies. ICML 2023: 18774-18795.